# Policy Space Diversity for Non-Transitive Games

**Jian Yao**[1*], **Weiming Liu** [1*], **Haobo Fu**[1*], **Yaodong Yang**[2],
**Stephen McAleer**[3], **Qiang Fu**[1], **Wei Yang**[1]
[1]Tencent AI Lab, Shenzhen, China
[2]Peking University, Beijing, China
[3]Carnegie Mellon University

## Abstract

Policy-Space Response Oracles (PSRO) is an influential algorithm framework for approximating a Nash Equilibrium (NE) in multi-agent non-transitive games. Many previous studies have been trying to promote policy diversity in PSRO. A major weakness in existing diversity metrics is that a more diverse (according to their diversity metrics) population does not necessarily mean (as we proved in the paper) a better approximation to a NE. To alleviate this problem, we propose a new diversity metric, the improvement of which guarantees a better approximation to a NE. Meanwhile, we develop a practical and well-justified method to optimize our diversity metric using only state-action samples. By incorporating our diversity regularization into the best response solving in PSRO, we obtain a new PSRO variant, *Policy Space Diversity* PSRO (PSD-PSRO). We present the convergence property of PSD-PSRO. Empirically, extensive experiments on various games demonstrate that PSD-PSRO is more effective in producing significantly less exploitable policies than state-of-the-art PSRO variants. The experiment code is available at https://github.com/nigelyaoj/policy-space-diversity-psro.

## 1 Introduction

Most real-world games demonstrate strong non-transitivity [9], where the winning rule follows a cyclic pattern (e.g., the strategy cycle in Rock-Paper-Scissors) [6, 3]. A common objective in solving non-transitive games is to find a Nash Equilibrium (NE), which has the best worst-case performance in the whole policy space. Traditional algorithms, like simple self-play, fail to converge to a NE in games with strong non-transitivity [26]. Recently, many game-theoretic methods have been proposed to approximate a NE in such games. For example, Counterfactual Regret Minimization (CFR) [54] minimizes the so-called counterfactual regret. Neural fictitious self play [18, 19] extends the classical game-theoretic approach, Fictitious Play (FP) [4], to larger games using Reinforcement Learning (RL) to approximate a Best Response (BR). Another well-known algorithm is Policy-Space Response Oracles (PSRO) [26], which generalizes the double oracle approach [38] by adopting a RL subroutine to approximate a BR.

Improving the performance of PSRO on approximating a NE is an active research topic, and many PSRO variants have been proposed so far, which generally fall into three categories. The first category [36, 45, 12, 29] aims to improve the training efficiency at each iteration. For instance, pipeline-PSRO [36] trains multiple BRs in parallel at each iteration. Neural population learning [29] enables fast transfer learning across policies via representing a population of policies within a single conditional model. The second category [37, 53] incorporates no-regret learning into PSRO, which solves an unrestricted-restricted game with a no-regret learning method to guarantee the decrease of *exploitability* of the meta-strategy across each iteration. The third category [2, 41, 32, 33]

---

*Equal contribution. Correspondence to: Haobo Fu (haobofu@tencent.com).

promotes policy diversity in the population, which is usually implemented by incorporating a diversity regularization term into the BR solving in the original PSRO.

Despite achieving promising improvements over the original PSRO, the theoretical reason why the diversity metrics in existing diversity-enhancing PSRO variants [2, 41, 32, 33] help PSRO in terms of approximating a NE is unclear. More specifically, those diversity metrics are 'justified' in the sense that adding the corresponding diversity-regularized BR strictly enlarges the *gamescape*. However, as we prove and demonstrate later in the paper, a population with a larger *gamescape* is neither a sufficient nor a necessary condition for a better approximation (we will define the precise meaning later) to a NE. One fundamental reason is that *gamescape* is a concept that varies significantly according to the choice of opponent policies. In contrast, *exploitability* (the distance to a NE) measures the worst-case performance that is invariant to the choice of opponent policies.

In this paper, we seek for a new and better justified diversity metric that improves the approximation of a NE in PSRO. To achieve this, we introduce a new concept, named *Population Exploitability* (PE), to quantify the strength of a population. The PE of a population is the optimal *exploitability* that can be achieved by selecting a policy from its *Policy Hull* (PH), which is simply a complete set of polices that are convex combinations of individual polices in the population. In addition, we show that a larger PH means a lower PE. Based on these insights, we make the following contributions:

- We point out a major and common weakness of existing diversity-enhancing PSRO variants: their goal of enlarging the *gamescape* of the population in PSRO is somewhat deceptive to the extent that it can lead to a weaker population in terms of PE. In other words, a more diverse (according to their diversity metrics) population $\Rightarrow$ a larger *gamescape* $\not\Rightarrow$ closer to a full game NE.

- We develop a new diversity metric that encourages the enlargement of a population's PH. In addition, we develop a practical and well-justified method to optimize our diversity metric using only state-action samples. We then incorporate our diversity metric (as a regularization term) into the BR solving in the original PSRO and obtain a new algorithm: Policy Space Diversity PSRO (PSD-PSRO). Our method PSD-PSRO establishes the causality: a more diverse (according to our diversity metric) population $\Rightarrow$ a larger PH $\Rightarrow$ a lower PE $\Rightarrow$ closer to a full game NE.

- We prove that a full game NE is guaranteed once PSD-PSRO is converged. In contrast, it is not clear, in other state-of-the-art diversity-enhancing PSRO variants [2, 41, 32, 33], whether a full game NE is found once they are converged in terms of their optimization objectives. Notably, $\text{PSRO}_{rN}$ [2] is not guaranteed to find a NE once converged [36].

## 2 Notations and Preliminary

### 2.1 Extensive-form Games, NE, and Exploitability

Extensive-form games are used to model sequential interaction involving multiple agents, which can be defined by a tuple $\langle \mathcal{N}, \mathcal{S}, P, \mathcal{A}, u \rangle$. $\mathcal{N} = \{1, 2\}$ denotes the set of players (we focus on the two-player zero-sum games). $\mathcal{S}$ is a set of information states for decision-making. Each information state node $s \in \mathcal{S}$ includes a set of actions $\mathcal{A}(s)$ that lead to subsequent information states. The player function $P : \mathcal{S} \to \mathcal{N} \cup \{c\}$, with $c$ denoting chance, determines which player takes action in $s$. We use $s_i, \mathcal{S}_i = \{s \in \mathcal{S} | P(s) = i\}$, and $\mathcal{A}_i = \cup_{s \in \mathcal{S}_i} \mathcal{A}(s)$ to denote player $i$'s state, set of states, and set of actions respectively. We consider games with *perfect recall*, where each player remembers the sequence of states to the current state.

A player's *behavioral strategy* is denoted by $\pi_i(s) \in \Delta(\mathcal{A}(s)), \forall s \in \mathcal{S}_i$, and $\pi_i(a|s)$ is the probability of player $i$ taking action $a$ in $s$. A *strategy profile* $\pi = (\pi_1, \pi_2)$ is a pair of strategies for each player, and we use $\pi_{-i}$ to refer to the strategy in $\pi$ except $\pi_i$. $u_i(\pi) = u_i(\pi_i, \pi_{-i})$ denotes the payoff for player $i$ when both players follow $\pi$. The BR of player $i$ to the opponent's strategy $\pi_{-i}$ is denoted by $\mathcal{BR}(\pi_{-i}) = \arg\max_{\pi'_i} u_i(\pi'_i, \pi_{-i})$. The *exploitability* of strategy profile $\pi$ is defined as:

$$\mathcal{E}(\pi) = \frac{1}{2} \sum_{i \in \mathcal{N}} [\max_{\pi'_i} u_i(\pi'_i, \pi_{-i}) - u_i(\pi_i, \pi_{-i})]. \tag{1}$$

When $\mathcal{E}(\pi) = 0$, $\pi$ is a NE of the game.

## 2.2 Meta-Games, PH, and PSRO

Meta-games are introduced to represent games at a higher level. Denoting a population of mixed strategies for player $i$ by $\Pi_i := \{\pi_i^1, \pi_i^2, ...\}$, the payoff matrix on the joint population $\Pi = \Pi_i \times \Pi_{-i}$ is denoted by $\mathbf{M}_{\Pi_i, \Pi_{-i}}$, where $\mathbf{M}_{\Pi_i, \Pi_{-i}}[j, k] := u_i(\pi_i^j, \pi_{-i}^k)$. The meta-game on $\Pi$ and $\mathbf{M}_{\Pi_i, \Pi_{-i}}$ is simply a normal-form game where selecting an action means choosing which $\pi_i$ to play for player $i$. Accordingly, we use $\sigma_i$ ($\sigma_i$ is called a meta-strategy and could be, e.g., playing $[\pi_i^1, \pi_i^2]$ with probability $[0.5, 0.5]$) to denote a mixed strategy over $\Pi_i$, i.e., $\sigma_i \in \Delta_{\Pi_i}$. A meta-policy $\sigma_i$ over $\Pi_i$ can be viewed as a convex combination of polices in $\Pi_i$, and we define the PH of a population $\mathcal{H}(\Pi_i)$ as the set of all convex combinations of the policies in $\Pi_i$. Meta-games are often open-ended in the sense that there exist an infinite number of mixed strategies and that new policies will be successively added to $\Pi_i$ and $\Pi_{-i}$ respectively. We give a summary of notations in Appendix A.

PSRO operates on meta-games and consists of two components: an oracle and a meta-policy solver. At each iteration $t$, PSRO maintains a population of policies, denoted by $\Pi_i^t$, for each player $i$. The joint meta-policy solver first computes a NE meta-policy $\sigma^t$ on the restricted meta-game represented by $\mathbf{M}_{\Pi_i^t, \Pi_{-i}^t}$. Afterwards, for each player $i$, the oracle computes an approximate BR (i.e., $\pi_i^{t+1}$) against the meta-policy $\sigma_{-i}^t$: $\pi_i^{t+1} \in \mathcal{BR}(\sigma_{-i}^t)$. The new policy $\pi_i^{t+1}$ is then added to its population ($\Pi_i^{t+1} = \Pi_i^t \cup \{\pi_i^{t+1}\}$), and the next iteration starts. In the end, PSRO outputs a meta-policy NE on the final joint population as an approximation to a full game NE.

## 2.3 Previous Diversity Metrics for PSRO

**Effective diversity** [2] measures the variety of effective strategies (strategies with support under a meta-policy NE) and uses a rectifier to focus on how these effective strategies beat each other. Let $(\sigma_i^*, \sigma_{-i}^*)$ denote a meta-policy NE on $\mathbf{M}_{\Pi_i, \Pi_{-i}}$. The *effective diversity* of $\Pi_i$ is:

$$\text{Div}(\Pi_i) = \sigma_i^{*T} \lfloor \mathbf{M}_{\Pi_i, \Pi_{-i}} \rfloor_+ \sigma_{-i}^*, \tag{2}$$

where $\lfloor x \rfloor_+ := x$ if x $\geq 0$ else 0.

**Expected Cardinality** [41], inspired by the determinantal point processes [35], measures the diversity of a population $\Pi_i$ as the expected cardinality of the random set $\mathbf{Y}$ sampled according to $det(\mathcal{L}_\Pi)$:

$$\text{Div}(\Pi_i) = \mathbb{E}_{\mathbf{Y} \sim \mathbb{P}_{\mathcal{L}_\Pi}}[|\mathbf{Y}|] = \text{Tr}(\mathbf{I} - (\mathcal{L}_\Pi + \mathbf{I})^{-1}), \tag{3}$$

where $|\mathbf{Y}|$ is the cardinality of $\mathbf{Y}$, and $\mathcal{L}_\Pi = \mathbf{M}_{\Pi_i, \Pi_{-i}} \mathbf{M}^T{}_{\Pi_i, \Pi_{-i}}$.

**Convex Hull Enlargement** [32] builds on the idea of enlarging the convex hull of all row vectors in the payoff matrix:

$$\text{Div}(\Pi_i \cup \{\pi_i'\}) = \min_{\mathbf{1}^T \beta = 1, \beta \geq 0} ||\mathbf{M}_{\Pi_i, \Pi_{-i}}^T \beta - \mathbf{m}||, \tag{4}$$

where $\pi_i'$ is the new strategy to add, and $\mathbf{m}$ is the payoff vector of policy $\pi_i'$ against each opponent policy in $\Pi_{-i}$: $\mathbf{m}[j] = u_i(\pi_i', \pi_{-i}^j)$.

**Occupancy Measure Mismatching** [32] considers the state-action distribution $\rho_\pi(s, a)$ induced by a joint policy $\pi$. When considering adding a new policy $\pi_i'$, the corresponding diversity metric is:

$$\text{Div}(\Pi_i \cup \{\pi_i'\}) = D_f(\rho_{(\pi_i', \sigma_{-i}^*)} || \rho_{(\sigma_i^*, \sigma_{-i}^*)}), \tag{5}$$

where $\pi_i'$ is the new policy to add; $(\sigma_i^*, \sigma_{-i}^*)$ is a meta-policy NE on $\mathbf{M}_{\Pi_i, \Pi_{-i}}$, and $D_f$ is a general $f$-divergence between two distributions. It is worth noting that Equation 5 only considers the difference between two policies ($\pi_i'$ and $\sigma_i^*$), instead of $\pi_i'$ and $\Pi_i$. In practice, this diversity metric is used together with the **convex hull enlargement** in [32].

**Unified Diversity Measure** [33] offers a unified view on existing diversity metrics and is defined as:

$$\text{Div}(\Pi_i) = \sum_{m=1}^{|\Pi_i|} f(\lambda_m), \tag{6}$$

where $f$ takes different forms for different existing diversity metrics; $\lambda_m$ is the eigenvalues of $[K(\phi_m, \phi_n)]_{|\Pi_i| \times |\Pi_i|}$; $K(\cdot, \cdot)$ is a predefined kernel function; and $\phi_m$ is the strategy feature for the $m$-th policy in $\Pi_i$. It is worth mentioning that only payoff vectors in $\mathbf{M}_{\Pi_i, \Pi_{-i}}$ were investigated for the strategy feature of the new diversity metric proposed in [33].

## 3 A Common Weakness of Existing Diversity-Enhancing PSRO Variants

As shown in last section, all previous diversity-enhancing PSRO variants [2, 41, 32, 33] try to enlarge the *gamescape* of $\Pi_i$, which is the convex hull of the rows in the empirical payoff matrix:

$$\mathcal{GS}(\Pi_i | \Pi_{-i}) := \{\sum_j \alpha_j \mathbf{m}_j : \boldsymbol{\alpha} \geq 0, \boldsymbol{\alpha}^T \mathbf{1} = 1\},$$

where $\mathbf{m}_j$ is the $j$-th row vector in $\mathbf{M}_{\Pi_i, \Pi_{-i}}$. However, the *gamescape* of a population depends on the choice of opponent policies, and two policies with the same payoff vector are not necessarily the same. Moreover, enlarging the *gamescape* without careful tuning would encourage the current player to deliberately lose to the opponent to get 'diverse' payoffs. We suspect this might be the reason why the optimization of the *gamescape* is activated later in the training procedure in [32]. More importantly, it is not theoretically clear from previous diversity-enhancing PSRO variants why enlarging the *gamescape* would help in approximating a full game NE in PSRO.

To rigorously answer the question whether a diversity metric is helpful in approximating a NE in PSRO, we need a performance measure to monitor the progress of PSRO across iterations in terms of finding a full game NE. In other words, we need to quantify the strength of a population of policies. Previously, the *exploitability* of a meta NE of the joint population is usually employed to monitor the progress of PSRO. Yet, as demonstrated in [37], this *exploitability* may increase after an iteration. Intuitively, a better alternative is the *exploitability* of the least exploitable mixed strategy supported by a population. We define this *exploitability* as the *population exploitability*:

**Definition 3.1.** For a joint population $\Pi = \Pi_i \times \Pi_{-i}$, let $(\sigma_i^*, \sigma_{-i}^*)$ be a meta NE on $\mathbf{M}_{\Pi_i, \Pi_{-i}}$. The *relative population performance* of $\Pi_i$ against $\Pi_{-i}$ [2] is:

$$\mathcal{P}_i(\Pi_i, \Pi_{-i}) = \sigma_i^{*T} \mathbf{M}_{\Pi_i, \Pi_{-i}} \sigma_{-i}^*. \tag{7}$$

The *population exploitability* of the joint population $\Pi$ is defined as:

$$\mathcal{PE}(\Pi) = \frac{1}{2} \sum_{i=1,2} \max_{\Pi_i' \subseteq \Omega_i} \mathcal{P}_i(\Pi_i', \Pi_{-i}), \tag{8}$$

where $\Omega_i$ is the full set of all possible mixed strategies of player $i$.

We notice that PE is equal to the sum of negative *population effectivity* defined in [32]. Yet, we prefer PE as it is more of a natural extension to *exploitability* in Equation 1. Some properties of PE and its relation to PH are presented in the following.

**Proposition 3.2.** *Considering a joint population* $\Pi = \Pi_i \times \Pi_{-i}$, *we have:*

1. $\mathcal{PE}(\Pi) \geq 0, \forall \Pi.$

2. *For another joint population* $\Pi'$, *if* $\mathcal{H}(\Pi_i) \times \mathcal{H}(\Pi_2) \subseteq \mathcal{H}(\Pi_i') \times \mathcal{H}(\Pi_{-i}')$, *then* $\mathcal{PE}(\Pi) \geq \mathcal{PE}(\Pi')$.

3. *If* $\Pi_i = \{\pi_i\}$ *and* $\Pi_{-i} = \{\pi_{-i}\}$, *then* $\mathcal{PE}(\Pi) = \mathcal{E}(\pi)$, *where* $\pi = (\pi_i, \pi_{-i})$.

4. $\exists \pi = (\pi_i, \pi_{-i}) \in \mathcal{H}(\Pi_i) \times \mathcal{H}(\Pi_{-i})$ *s.t.* $\mathcal{E}(\pi) = \mathcal{PE}(\Pi) = \min_{\pi' \in \mathcal{H}(\Pi_i) \times \mathcal{H}(\Pi_{-i})} \mathcal{E}(\pi')$.

5. *Let* $(\sigma_i^*, \sigma_{-i}^*)$ *denote an arbitrary NE of the full game.* $\mathcal{PE}(\Pi) = 0$ *if and only if* $(\sigma_i^*, \sigma_{-i}^*) \in \mathcal{H}(\Pi_i) \times \mathcal{H}(\Pi_{-i})$.

The proof is in Appendix B.2. Once we use PE to monitor the progress of PSRO, we have:

**Proposition 3.3.** *The PE of the joint population* $\Pi^t$ *at each iteration* $t$ *in PSRO is monotonically decreasing and will converge to 0 in finite iterations for finite games. Once* $\mathcal{PE}(\Pi^T) = 0$, *a meta NE on* $\Pi^T$ *is a full game NE.*

The proof is in Appendix B.3. From Proposition 3.2 and 3.3, we are convinced that PE is indeed an appropriate performance measure for populations of polices. Using PE, we can now formally present why enlarging the *gamescape* of the population in PSRO is somewhat deceptive:

**Theorem 3.4.** *The enlargement of the gamescape is neither sufficient nor necessary for the decrease of PE. Considering two populations* ($\Pi_i^1$ *and* $\Pi_i^2$) *for player* $i$ *and one population* $\Pi_{-i}$ *for player* $-i$, *and denoting* $\Pi^j = \Pi_i^j \times \Pi_{-i}$, $j = 1, 2$ *we have*

$$\mathcal{GS}(\Pi_i^1 | \Pi_{-i}) \subsetneq \mathcal{GS}(\Pi_i^2 | \Pi_{-i}) \not\Rightarrow \mathcal{PE}(\Pi^1) \geq \mathcal{PE}(\Pi^2)$$
$$\mathcal{PE}(\Pi^1) \geq \mathcal{PE}(\Pi^2) \not\Rightarrow \mathcal{GS}(\Pi_i^1 | \Pi_{-i}) \subsetneq \mathcal{GS}(\Pi_i^2 | \Pi_{-i}) \tag{9}$$

The proof of Theorem 3.4 is in Appendix B.5, where we provide concrete examples.

In other words, enlarging the gamescape in either short term or long term does not necessarily lead to a better approximation to a full game NE.

# 4 Policy Space Diversity PSRO

In this section, we develop a new diversity-enhancing PSRO variant, i.e., PSD-PSRO. In contrast to methods that enlarge the *gamescape*, PSD-PSRO encourages the enlargement of PH of a population, which helps reduce a population's PE (according to Proposition 3.2). In addition, we develop a well-justified state-action sampling method to optimize our diversity metric in practice. Finally, we present the convergence property of PSD-PSRO and discuss its relation to the original PSRO.

## 4.1 A New Diversity Regularization Term for PSRO

Our purpose of promoting diversity in PSRO is to facilitate the convergence to a full game NE. We follow the conventional scheme in previous diversity-enhancing PSRO variants [41, 32, 33], which introduces a diversity regularization term to the BR solving in PSRO. Nonetheless, our diversity regularization encourages the enlargement of PH of the current population, which is in contrast to the enlargement of *gamescape* in previous methods [2, 41, 32, 33]. We thus name our diversity metric *policy space diversity*. Intuitively, the larger the PH of a population is, the more likely it will include a full game NE. More formally, a larger PH means a lower PE (Proposition 3.2), which means our diversity metric avoids the common weakness (Section 3) of existing ones.

Recall that the PH of a population is simply the complete set of polices that are convex combinations of individual polices in the population. To quantify the contribution of a new policy to the enlargement of the PH of the current population, a straightforward idea is to maximize a distance between the new policy and the PH. Such a distance should be 0 for any policy that belongs to the PH and greater than 0 otherwise. Without loss of generality, the distance between a policy and a PH could be defined as the minimal distance between the policy and any policy in the PH. We can now write down the diversity regularized BR solving objective in PSD-PSRO, where at each iteration $t$ for player $i$ we add a new policy $\pi_i^{t+1}$ by solving:

$$\pi_i^{t+1} = \arg\max_{\pi_i} \left\{ u(\pi_i, \sigma_{-i}^t) + \lambda \min_{\pi_i^k \in \mathcal{H}(\Pi_i^t)} \text{dist}(\pi_i, \pi_i^k) \right\}, \tag{10}$$

where $\sigma_{-i}^t$ is the opponent's meta NE policy at the $t$-th iteration, $\lambda$ is a Lagrange multiplier, and $\text{dist}(\cdot, \cdot)$ is a distance function (will be specified in the next subsection) between two polices.

## 4.2 Diversity Optimization in Practice

To be able to optimize our diversity metric (the right part in Equation 10) in practice, we need to encode a policy into some representation space and specify a distance function there. Such a representation should be a one-to-one mapping between a policy and its representation. Also, to ensure that enlarging the convex hull in the representation space results in the enlargement of the PH, we require the representation to satisfy the linearity property. Formally, we have the following definition:

**Definition 4.1.** A *fine policy representation* for our purpose is a function $\rho : \Pi_i \to R^{N_i}$, which satisfies the following two properties:

- (bijection) For any representation $\rho(\pi_i)$, there exists a unique behavior policy $\pi_i$ whose representation is $\rho(\pi_i)$, and vice-versa.

- (linearity) For any two policies ($\pi_i^j$ and $\pi_i^k$) and $\alpha$ ($0 \leq \alpha \leq 1$), the following holds:

$$\rho(\alpha\pi_i^j + (1-\alpha)\pi_i^k) = \alpha\rho(\pi_i^j) + (1-\alpha)\rho(\pi_i^k),$$

where $\alpha\pi_i^j + (1-\alpha)\pi_i^k$ means playing $\pi_i^j$ with probability $\alpha$ and $\pi_i^k$ with probability $(1-\alpha)$.

Existing diversity metrics explicitly or implicitly define a policy representation [33]. For instance, the *gamescape*-based methods [2, 41, 32, 33] represent a policy using its payoff vector against the opponent's population. Yet, this representation is not a *fine policy representation* as it is not a bijection (different policies can have the same payoff vector). The (joint) occupancy measure, which is a *fine policy representation*, is usually used to encode a policy in the RL community [48, 20, 32]. The $f$-divergence is then employed to measure the distance between two policies [32, 24, 16]. However, computing the $f$-divergence based on the occupancy measure is usually intractable and often in practice roughly approximated using the prediction of neural networks [32, 24, 16].

Instead, we use another *fine policy representation*, i.e., the sequence-form representation [21, 11, 31, 27, 1], which was originally developed for representing a policy in multi-agent games. We then define the distance between two policies using the Bregman divergence, which can be further simplified to a tractable form and optimized using only state-action samples in practice.

The sequence-form representation of a policy remembers the realization probability of reaching a state-action pair. We follow the definition in [11, 31], where the sequence form representation $\boldsymbol{x}_i \in \mathcal{X}_i \subseteq [0,1]^{|\mathcal{S}_i \times \mathcal{A}_i|}$ of $\pi_i$ is a vector:

$$\boldsymbol{x}_i(s,a) = \prod_{\widetilde{s}_i, \widetilde{a} \in \tau(s,a)} \pi_i(\widetilde{a}|\widetilde{s}_i), \tag{11}$$

where $\tau(s,a)$ is a trajectory from the beginning to $(s,a)$. By the perfect-recall assumption, there is a unique $\tau$ that leads to $(s,a)$. The policy $\pi_i$ can be written as $\pi_i(a|s) = \boldsymbol{x}_i(s,a)/\|\boldsymbol{x}_i(s)\|_1$, where $\boldsymbol{x}_i(s)$ is $(\boldsymbol{x}_i(s,a_1), \ldots, \boldsymbol{x}_i(s,a_n))$ with $a_1, \ldots, a_n \in \mathcal{A}(s)$. Unlike the payoff vector representation or the occupancy measure representation, $\boldsymbol{x}_i$ is independent of the opponent's policy as well as the environmental dynamics. Therefore, it should be more appropriate in representing a policy for the diversity optimization. Without loss of generality and following [21, 11, 31], we define the distance $\text{dist}(\pi_i, \pi_i')$ between two policies as the Bregman divergence on the sequence form representation $\mathcal{B}_d(\boldsymbol{x}_i\|\boldsymbol{x}_i')$, which can be further written in terms of state-action pairs (the derivation is presented in Appendix B.1) in the following:

$$\text{dist}(\pi_i, \pi_i') := \mathcal{B}_d(\boldsymbol{x}_i\|\boldsymbol{x}_i') = \sum_{s,a \in \mathcal{S}_i \times \mathcal{A}_i} \left( \prod_{\widetilde{s}_i, \widetilde{a} \in \tau(s,a)} \pi_i(\widetilde{a}|\widetilde{s}_i) \right) \beta_s \mathcal{B}_{d_s}(\pi_i(s)\|\pi_i'(s)), \tag{12}$$

where $\mathcal{B}_{d_s}(\pi_i(s)\|\pi_i'(s))$ is the Bregman divergence between $\pi_i(s)$ and $\pi_i'(s)$. In our experiment, we let $\mathcal{B}_{d_s}(\pi_i(s)\|\pi_i'(s)) = \sum_a \pi_i(a|s) \log \pi_i(a|s)/\pi_i'(a|s) = \text{KL}(\pi_i(s)\|\pi_i'(s))$, i.e., the KL divergence. In previous work [21, 11, 31], the coefficient $\beta_s$ usually declines monotonically as the length of the sequence increases. In our case, we make $\beta_s$ depend on an opponent's policy $b_{-i}$: $\beta_s = \prod_{\widetilde{s}_{-i}, \widetilde{a} \in \tau(s,a)} b_{-i}(\widetilde{a}|\widetilde{s}_{-i})^2$. This weighting method allows us to estimate the distance using the sampled average KL divergence and avoids importance sampling:

$$\begin{aligned} \text{dist}(\pi_i, \pi_i') &= \sum_{s,a \in \mathcal{S}_i \times \mathcal{A}_i} \left( \prod_{\widetilde{s}_i, \widetilde{a} \in \tau(s,a)} \pi_i(\widetilde{a}|\widetilde{s}_i) \right) \left( \prod_{\widetilde{s}_{-i}, \widetilde{a} \in \tau(s,a)} b_{-i}(\widetilde{a}|\widetilde{s}_{-i}) \right) \mathcal{B}_{d_s}(\pi_i(s)\|\pi_i'(s)) \\ &= \mathbb{E}_{s_i \sim \pi_i, b_{-i}}[\text{KL}(\pi_i(s_i)\|\pi_i'(s_i))], \end{aligned} \tag{13}$$

where $s_i \sim \pi_i, b_{-i}$ means sampling player $i$'s information states from the trajectories that are collected by playing $\pi_i$ against $b_{-i}$. As a result, we can rewrite the diversity regularized BR solving objective in PSD-PSRO as follows:

$$\pi_i^{t+1} = \arg\max_{\pi_i} \left\{ u(\pi_i, \sigma_{-i}^t) + \lambda \min_{\pi_i^k \in \mathcal{H}(\Pi_i^t)} \mathbb{E}_{s_i \sim \pi_i, b_{-i}}[\text{KL}(\pi_i(s_i)\|\pi_i^k(s_i))] \right\}. \tag{14}$$

Now we provide a practical way to optimize Equation 14. By regarding the opponent and chance as the environment, we can use the policy gradient method [47, 43, 44] in RL to train $\pi_i^{t+1}$. Denote the probability of generating $\tau$ by $\pi_i(\tau)$, and the payoff of player $i$ for the trajectory $\tau$ by $R(\tau)$. Let $\pi_i^{min} = \arg\min_{\pi_i^k \in \mathcal{H}(\Pi_i^t)} \mathbb{E}_{s_i \sim \pi_i, b_{-i}}[\text{KL}(\pi_i(s_i)\|\pi_i^k(s_i))]$ be the policy in $\mathcal{H}(\Pi_i^t)$ that minimizes the distance and let $R^{KL}(\tau) = \mathbb{E}_{s_i \sim \tau}[\text{KL}(\pi_i(s_i)\|\pi_i^{min}(s_i))]$.

---

[2] Assume $\beta_s > 0$, i.e., $b_{-i}(a|s_{-i}) > 0, \forall(s_{-i}, a) \in \mathcal{S}_{-i} \times \mathcal{A}_{-i}$.

The gradient of the first term $u(\pi_i, \sigma_{-i}^t)$ in Equation 14 with respect to $\pi_i$ can be written as:

$$\nabla u(\pi_i, \sigma_{-i}^t) = \nabla \int \pi_i(\tau) R(\tau) \mathrm{d}\tau = \int \nabla \pi_i(\tau) \mathrm{R}(\tau) \mathrm{d}\tau$$

$$= \int \pi_i(\tau) \nabla \log \pi_i(\tau) R(\tau) \mathrm{d}\tau$$

$$= \mathbb{E}_{\tau \sim \pi_i, \sigma_{-i}^t} [\nabla \log \pi_i(\tau) R(\tau)]. \tag{15}$$

The gradient of the diversity term in Equation 14 with respect to $\pi_i$ can be written as:

$$\nabla_{\pi_i} \min_{\pi_i^k \in \mathcal{H}(\Pi_i^t)} \mathbb{E}_{s_i \in \tau, \tau \sim \pi_i, b_{-i}} [\mathrm{KL}(\pi_i(s_i) \| \pi_i^k(s_i))]$$

$$= \nabla \mathbb{E}_{s_i \in \tau, \tau \sim \pi_i, b_{-i}} [\mathrm{KL}(\pi_i(s_i) \| \pi_i^{min}(s_i))]$$

$$= \nabla \int \pi_i(\tau) R^{KL}(\tau) \mathrm{d}\tau$$

$$= \int \nabla \pi_i(\tau) R^{KL}(\tau) \mathrm{d}\tau + \int \pi_i(\tau) \mathbb{E}_{s_i \sim \tau} [\nabla \mathrm{KL}(\pi_i(s_i) \| \pi_i^{min}(s_i))] \mathrm{d}\tau$$

$$= \mathbb{E}_{\tau \sim \pi_i, b_{-i}} [\nabla \log \pi_i(\tau) R^{KL}(\tau)] + \mathbb{E}_{s_i \sim \pi_i, b_{-i}} [\nabla \mathrm{KL}(\pi_i(s_i) \| \pi_i^{min}(s_i))], \tag{16}$$

where we use the property that $\nabla \mathbb{E}[KL(\pi_i(s_i) | \pi_i^{min}(s_i)] \in \partial \min_{\pi_i^k} \mathbb{E}[KL(\pi_i(s_i) | \pi_i^k(s_i)]$, whose correctness can be shown from the following proposition:

**Proposition 4.2.** *For any local Lipschitz continuous function $f(x, y)$, assume $\forall x$, $\min_y f(x, y)$ exists, then $\partial_x f(x, y)|_{y \in \arg \min f(x,y)} \in \partial_x \min_y f(x, y)$, where $\partial f$ is the generalized gradient [7].*

*Proof.* According to Theorem 2.1 (property (4) in [7]), the result is immediate, as $\partial_x \min_y f(x, y)$ is the convex hull of $\{\partial_x f(x, y) | y \in \arg \max g(x, y)\}$. □

Combining the above two equations, we have,

$$\nabla_{\pi_i} \left( u(\pi_i, \sigma_{-i}^t) + \lambda \min_{\pi_i^k \in \mathcal{H}(\Pi_i^t)} \mathbb{E}_{s_i \sim \pi_i, b_{-i}} [\mathrm{KL}(\pi_i(s_i) \| \pi_i^k(s_i))] \right)$$

$$= \mathbb{E}_{\tau \sim \pi_i, \sigma_{-i}^t} [\nabla \log \pi_i(\tau) R(\tau)] + \lambda \mathbb{E}_{\tau \sim \pi_i, b_{-i}} [\nabla \log \pi_i(\tau) R^{KL}(\tau)]$$

$$+ \lambda \mathbb{E}_{s_i \sim \pi_i, b_{-i}} [\nabla \mathrm{KL}(\pi_i(s_i) \| \pi_i^{min}(s_i))]. \tag{17}$$

According to Equation 17, we can see that optimizing $\pi_i^{t+1}$ requires maximizing two types of rewards: $R(\tau)$ and $R^{KL}(\tau)$. This can be done by averaging the gradients using samples that are generated by playing $\pi_i$ against $\sigma_{-i}^t$ and $b_{-i}$ separately. The last term in Equation 17 is easily estimated by sampling states in the sampled trajectories via playing $\pi_i$ against $b_{-i}$. For training efficiency, we simply set $b_{-i} = \sigma_{-i}^t$ in our experiments, although other settings of $b_{-i}$ are possible, e.g., the uniform random policy. We also want to emphasize that we optimize $\pi_i^{t+1}$ for each iteration $t$, and $\sigma_{-i}^t$ is fixed during an iteration and thus can be viewed as a part of the environment. Finally, to estimate the distance between $\pi_i$ and $\pi_i^{min}$, we should ideally compute $\pi_i^{min}$ exactly first by solving a convex optimization problem. In practice, we find sampling the policies in $\mathcal{H}(\Pi_i^t)$ and using the minimal sampled distance already gives us satisfactory performance. The pseudo-code of PSD-PSRO is provided in Appendix C.

## 4.3 The Convergence Property of PSD-PSRO

We first show how the PH evolves in the original PSRO:

**Proposition 4.3.** *Before the PE of the joint population reaches 0, adding a BR to the meta NE policy from last iteration in PSRO will strictly enlarge the PH of the current population.*

The proof is in Appendix B.3. From Proposition 4.3, we can see that adding a BR serves one way (an implicit way) of enlarging the PH and hence reducing the PE. In contrast, the optimization of our diversity term in Equation 14 aims to explicitly enlarge the PH. In other words, adding a diversity regularized BR in PSD-PSRO serves as a mixed way of enlarging the PH. More formally, we have the following theorem:

**Theorem 4.4.** *(1) In PSD-PSRO, before the PE of the joint population reaches* 0*, adding an optimal solution in Equation 14 will strictly enlarge the PH and hence reduce the PE. (2) Once the PH can not be enlarged (i.e., PSD-PSRO converges) by adding an optimal solution in Equation 14, the PE reaches 0, and PSD-PSRO finds a full game NE.*

The proof is in Appendix B.6. In terms of convergence property, one significant benefit of PSD-PSRO over other state-of-the-art diversity-enhancing PSRO variants [2, 41, 32, 33] is the convergence of PH guarantees a full game NE in PSD-PSRO. Yet, it is not clear in those papers whether a full game NE is found once the PH of their populations converge. Notably, PSRO$_{rN}$ [2] is not guaranteed to find a NE once converged [36]. In practice, we expect a significant performance improvement of PSD-PSRO over PSRO in approximating a NE. As for different games, there might exist different optimal trade-offs between 'exploitation' (adding a BR) and 'exploration' (optimizing our diversity metric) in enlarging the PH and reducing the PE. In other words, PSD-PSRO generalizes PSRO (a PSD-PSRO instance when $\lambda = 0$) in ways of enlarging the PH and reducing the PE.

# 5   Related Work

Diversity has been widely studied in evolutionary computation [13], with a central focus that mimics the natural evolution process. One of the ideas is novelty search [28], which searches for policies that lead to novel outcomes. By hybridizing novelty search with a fitness objective, quality-diversity [42] aims for diverse behaviors of good qualities. Despite these methods achieving good empirical results [8, 23], the diversity metric is often hand-crafted for different tasks.

Promoting diversity is also intensively studied in RL. By adding a distance regularization between the current policy and a previous policy, a diversity-driven approach has been proposed for good exploration [22]. Unsupervised learning of diverse policies [10] has been studied to serve as an effective pretraining mechanism for downstream RL tasks. A diversity metric based on DPP [39] has been proposed to improve exploration in population-based training. Diverse behaviors were learned in order to improve generalization ability for test environments that are different from training [25]. A diversity-regularized collaborative exploration strategy has been proposed in [40]. Reward randomization [49] has been employed to discover diverse strategies in multi-agent games. Trajectory diversity has been studied for better zero-shot coordination in a multi-agent environment [34]. Quality-similar diversity has been investigated in [51].

Diversity also plays a role in game-theoretic methods. Smooth FP [17] adds a policy entropy term when finding a BR. PSRO$_{rN}$ [2] encourages effective diversity, which considers amplifying the strength over the weakness in a policy. DPP-PSRO [41] introduces a diversity metric based on DPP and provides a geometric interpretation of behavioral diversity. BD&RD-PSRO [32] combines the occupancy measure mismatch and the diversity on payoff vectors as a unified diversity metric. UDM [33] summarizes existing diversity metrics, by providing a unified diversity framework. In both opponent modeling [15] and opponent-limited subgame solving [30], diversity has been shown to have a large impact on the performance.

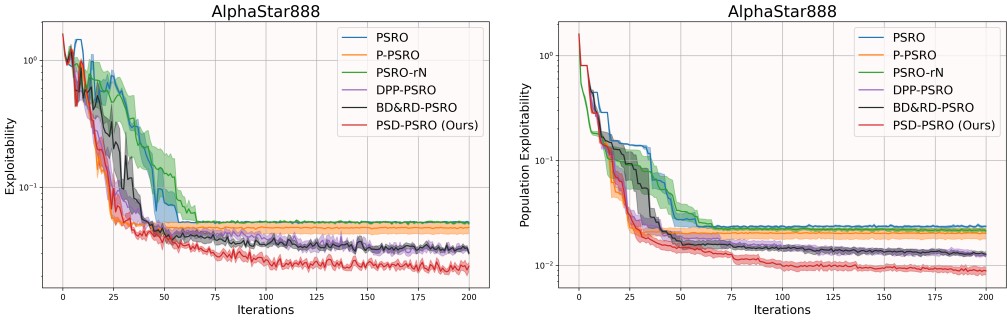

Figure 1:   (a): *Exploitability* of the meta NE. (b): PE of the joint population.

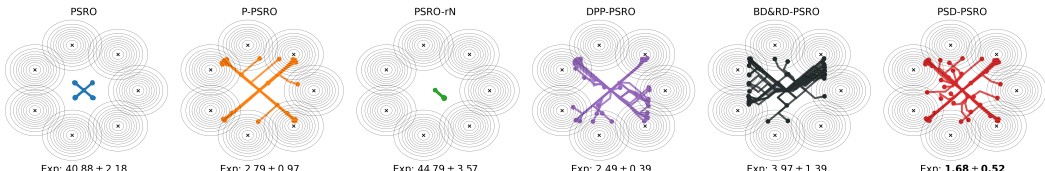

| PSRO | P-PSRO | PSRO-rN | DPP-PSRO | BD&RD-PSRO | PSD-PSRO |
|---|---|---|---|---|---|
| Exp: 40.88 ± 2.18 | Exp: 2.79 ± 0.97 | Exp: 44.79 ± 3.57 | Exp: 2.49 ± 0.39 | Exp: 3.97 ± 1.39 | Exp: **1.68 ± 0.52** |

Figure 2: Non-Transitive Mixture Game. Exploration trajectories during training. For each method, the final *exploitability* $\times 100$ (Exp) is reported at the bottom.

# 6 Experiments

The main purpose of the experiments is to compare PSD-PSRO with existing state-of-the-art PSRO variants in terms of approximating a full game NE. The baseline methods include PSRO [26], Pipeline-PSRO (P-PSRO) [36], PSRO$_{rN}$ [2], DPP-PSRO [41], and BD&RD-PSRO [32]. The benchmarks consist of single-state games (AlphaStar888 and non-transitive mixture game) and complex extensive games (Leduc poker and Goofspiel). For AlphaStar888 and Leduc poker, we report the *exploitability* of the meta NE and the PE of the joint population through the training process. For the non-transitive mixture game, we illustrate the 'diversity' of the population and report the final *exploitability*. For Goofspiel where the exact *exploitability* is intractable, we report the win rate between the final agents. In addition, we illustrate how the PH evolves for each method using the Disc game [2] in Appendix D.3, where PSD-PSRO is more effective at enlarging the PH and approximating a NE. An ablation study on $\lambda$ of PSD-PSRO in Appendix D.2 reveals that, for different benchmarks, an optimal trade-off between 'exploitation' and 'exploration' in enlarging the PH to approximate a NE usually happens when $\lambda$ is greater than zero. Error bars or stds in the results are obtained via 5 independent runs. In Appendix D.3, we also investigate the time cost of calculating our policy space diversity. More details for the environments and hyper-parameters are given in Appendix E.

**AlphaStar888** is an empirical game generated from the process of solving Starcraft II [50], which contains a payoff table for 888 RL policies. be viewed as a zero-sum symmetric two-player game where there is only one state $s_0$. In $s_0$, there are 888 legal actions. Any mixed strategy is a discrete probability distribution over the 888 actions. Hence, the distance function in Equation 14 for AlphaStar888 reduces to the KL divergence between two 888-dim discrete probability distributions.

In Figure 1, we can see that PSD-PSRO is more effective at reducing both the *exploitability* and PE than other methods.

**Non-Transitive Mixture Game** consists of 7 equally-distanced Gaussian humps on the 2D plane. Each strategy can be represented by a point on the 2D plane, which is equivalent to the weights (the likelihood of that point in each Gaussian distribution) that each player puts on the humps. The optimal strategy is to stay close to the center of the Gaussian humps and explore all the distributions. In Figure 2, we show the exploration trajectories for different methods during training, where PSRO and PSRO$_{rN}$ get trapped and fail in this non-transitive game. In contrast, PSD-PSRO tends to find diverse strategies and explore all Gaussians. Also, the *exploitability* of the meta NE of the final population is significantly lower in PSD-PSRO than others.

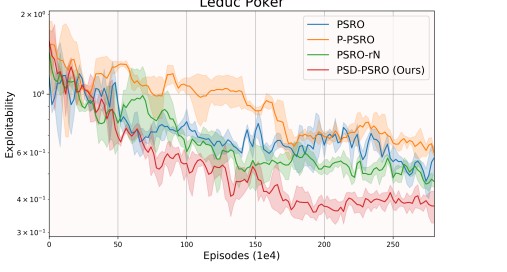 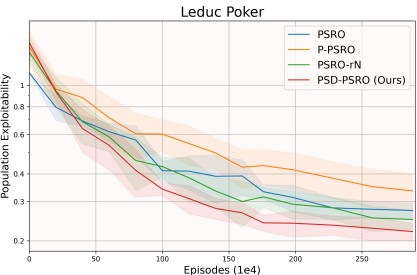

Figure 3: (a): *Exploitability* of the meta NE. (b): PE of the joint population.

**Leduc Poker** is a simplified poker [46], where the deck consists of two suits with three cards in each suit. Each player bets one chip as an ante, and a single private card is dealt to each player. Since DPP-PSRO cannot scale to the RL setting [32] and the code of BD&RD-PSRO for complex games is not available, we compare PSD-PSRO only to P-PSRO, $PSRO_{rN}$ and PSRO. As demonstrated in Figure 3, PSD-PSRO is more effective at reducing both the *exploitability* and PE.

|  | PSRO | $PSRO_{rN}$ | P-PSRO | PSD-PSRO(OURS) |
|---|---|---|---|---|
| PSRO | - | 0.613±0.019 | 0.469±0.034 | 0.422±0.025 |
| $PSRO_{rN}$ | 0.387±0.019 | - | 0.412±0.030 | 0.358±0.019 |
| P-PSRO | 0.531±0.034 | 0.588±0.030 | - | 0.370±0.031 |
| PSD-PSRO(OURS) | **0.578±0.025** | **0.642±0.019** | **0.630±0.031** | - |

Table 1: The win rate of the row agents against the column agents on Goofspiel.

**Goofspiel** is commonly used as a large-scale multi-stage simultaneous move game. Goofspiel features strong non-transitivity, as every pure strategy can be exploited by a simple counter-strategy. We compare PSD-PSRO with PSRO, P-PSRO, and $PSRO_{rN}$ on Goofspiel with 5 point cards and 8 point cards settings. In the game with 5 point cards setting, due to the relatively small game size, we can calculate the exact *exploitability*. We report the results in Appendix D.1, in which we see that PSD-PSRO reduces the *exploitability* more effectively than other methods. In the game with 8 point cards setting, the game size is too large to show exact *exploitability* for each iteration. In this setting, we provide a comparison among final solutions produced by different methods. We report the win rate between each two methods in Table 1, where we can see that PSD-PSRO consistently beats existing methods with a 62% win rate on average.

## 7  Conclusions and Limitations

In this paper, we point out a major and common weakness of existing diversity metrics in previous diversity-enhancing PSRO variants, which is their goal of enlarging the *gamescape* does not necessarily result in a better approximation to a full game NE. Based on the insight that a larger PH means a lower PE (a better approximation to a NE), we develop a new diversity metric (*policy space diversity*) that explicitly encourages the enlargement of a population's PH. We then develop a practical method to optimize our diversity metric using only state-action samples, which is derived based on the Bregman divergence on the sequence form of policies. We incorporate our diversity metric into the BR solving in PSRO to obtain PSD-PSRO. We present the convergence property of PSD-PSRO, and extensive experiments demonstrate that PSD-PSRO is significantly more effective in approximating a NE than state-of-the-art PSRO variants.

The diversity regularization term $\lambda$ in PSD-PSRO plays an important role in balancing the 'exploitation' and 'exploration' in terms of enlarging the PH to approximate a NE. In this paper, other than showing that different problems have different optimal settings of $\lambda$, we have not discussed about the guidance of choosing $\lambda$ optimally. Although there has been related work on how to adapt a diversity regularization term using online bandits [39], future work is still needed on how our *policy space diversity* could benefit the approximation of a full game NE most. Another interesting direction of future work is to extend PSD-PSRO to larger scale games, such as poker [5], Mahjong [14], and dark chess [52].

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

# A  Notation

We provide the notation in Table 2.

Table 2: Notations

| Notation | Meaning |
|---:|---|
| | **Extensive-Form Games** |
| $\mathcal{N}$ | $\mathcal{N} = \{1, 2\}$; set of players |
| $\mathcal{S}$ | set of information states |
| $s$ | $s \in \mathcal{S}$; state node |
| $\mathcal{A}(s)$ | set of actions allowed to be performed at state $s$ |
| $P$ | player function |
| $i \, / -i$ | $i \in \mathcal{N}$; player $i$ / players except $i$ |
| $s_i$ | player $i$'s information state |
| $\mathcal{S}_i$ | $\mathcal{S}_i = \{s \in \mathcal{S} | P(s) = i\}$; set of player's information state |
| $\mathcal{A}_i$ | $\mathcal{A}_i = \cup_{s \in \mathcal{S}_i} \mathcal{A}(s)$; set of player $i$'s actions |
| $\pi_i$ | player $i$'s behaivoral strategy |
| $\pi$ | $\pi = (\pi_i, \pi_{-i})$ strategy profile |
| $u_i(\pi)$ | $u_i(\pi) = u_i(\pi_i, \pi_{-i})$; the payoff for player $i$ |
| $\mathcal{BR}(\pi_{-i})$ | $\mathcal{BR}(\pi_{-i}) := \arg \max_{\pi_i' \in \Delta S_i} u(\pi_i', \pi_{-i})$; best response for player $i$ against players $-i$ |
| $\mathcal{E}(\pi)$ | $\mathcal{E}(\pi) = \frac{1}{N} \sum_{i \in \mathcal{N}} [\max_{\pi_i'} u_i(\pi_i', \pi_{-i}) - u_i(\pi_i, \pi_{-i})]$; exploitability for policy $\pi$ |
| | **Meta Games** |
| $i \, / -i$ | $i \in \{1, 2\}$; player $i$ / players except $i$ |
| $\pi_i^j \, / \, \pi_i$ | $j$-th policy / any policy for player $i$ |
| $\pi$ | $\pi = (\pi_1, \pi_2)$; joint policy |
| $\Pi_i$ | $\Pi_i := \{\pi_i^1, \pi_i^2, ..., \}$; policy set for player $i$ |
| $\Pi$ | $\Pi = \Pi_1 \times \Pi_2$; joint policy set |
| $\sigma_i$ | $\sigma_i \in \Delta_{\Pi_i}$; meta-policy over $\Pi_i$ |
| $\sigma$ | $\sigma = (\sigma_1, \sigma_2)$; joint meta-policy |
| $\mathbf{M}_{\Pi_i, \Pi_{-i}}$ | $(\mathbf{M}_{\Pi_i, \Pi_{-i}})_{jk} := u_i(\pi_i^j, \pi_{-i}^k)$; player $i$'s payoff table in the meta-games $(\Pi_i, \Pi_{-i})$ |
| $\Omega_i$ | the full set of all possible mixed strategies of player $i$ |
| $\mathcal{H}(\Pi_i)$ | $\mathcal{H}(\Pi_i) = \{\sum_j \beta_j \pi_i^j | \beta_j \geq 0, \sum_{j=1}^t \beta_j = 1\}$; Policy Hull of $\Pi_i = \{\pi_i^1, \pi_i^2, ..., \pi_i^t\}$ |
| $\mathcal{P}_i(\cdot, \cdot)$ | $\mathcal{P}_i(\Pi_i, \Pi_{-i}) = \pi_i^{*T} \mathbf{M}_{\Pi_i, \Pi_{-i}} \pi_{-i}^*$ ; Relative Population Performance for player $i$ |
| $\mathcal{PE}(\cdot)$ | $\mathcal{PE}(\Pi) = \frac{1}{2} \sum_{i=1,2} \max_{\Pi_i' \subseteq \Omega_i} \mathcal{P}_i(\Pi_i', \Pi_{-i})$; Population Exploitability |
| $\mathcal{GS}(\Pi_i | \Pi_{-i})$ | $\mathcal{GS}(\Pi_i | \Pi_{-i}) = \{\sum_j \alpha_j \mathbf{m}_j : \boldsymbol{\alpha} \geq 0, \boldsymbol{\alpha}^T \mathbf{1} = 1, \mathbf{m}_j = \mathbf{M}_{\Pi_i, \Pi_{-i}[j,:]}\}$; gamescape of $\Pi_i$ |
| | **PSRO / PSD-PSRO Training** |
| $\Pi_i^t$ | $\Pi_i^t = \{\pi_i^1, ..., \pi_i^t\}$; the policy set of player $i$ at the $t$-th iteration in PSRO |
| $\sigma_i^t$ | $\sigma_i^t \in \Delta_{\Pi_i^t}$; the meta-policy of player $i$ at $t$-th iteration in PSRO |
| $\rho(\cdot)$ | policy representation |
| $\boldsymbol{x}_i$ | $\boldsymbol{x}_i \in \mathcal{X}_i \subseteq [0, 1]^{|\mathcal{S}_i \times \mathcal{A}_i|}$ sequence form representation for a policy |
| $\tau(s, a)$ | trajectory from the initial state to $(s, a)$ |
| $\mathcal{B}_{d_s}(\pi_i(s) \| \pi_i'(s))$ | Bregman divergence between $\pi_i(s)$ and $\pi_i'(s)$ |
| $\text{Div}(\cdot)$ | diversity on the policy set specified in different methods |
| $\text{dist}(\cdot, \cdot)$ | distance function |
| $\lambda$ | diversity weight |
| KL | Kullback-Leibler divergence |

# B  Theoretical Analysis

## B.1  Derivation of Bregman Divergence

We define the Bregman divergence on the sequence-form representation of the policies. Given a strictly convex function $d$ defined on $\mathbb{R}^{|\mathcal{S}_i \times \mathcal{A}_i|}$, a Bregman divergence is defined as

$$\mathcal{B}_d(\boldsymbol{x}_i \| \boldsymbol{x}_i') = d(\boldsymbol{x}_i) - d(\boldsymbol{x}_i') - \langle \nabla d(\boldsymbol{x}_i'), \boldsymbol{x}_i - \boldsymbol{x}_i' \rangle, \tag{18}$$

for any $\boldsymbol{x}_i, \boldsymbol{x}_i' \in \mathcal{X}_i$. Intuitively, Bregman divergence is the gap between $d(\boldsymbol{x}_i)$ and its first-order approximation at $\boldsymbol{x}_i'$. It is non-negative (zero is achieved if and only if $\boldsymbol{x}_i = \boldsymbol{x}_i'$) and strictly convex in its first argument. So, it can be used to measure the difference between $\boldsymbol{x}_i$ and $\boldsymbol{x}_i'$. Examples of Bregman divergence include the Squared Euclidean distance, which is generated by the $l_2$ function, and the Kullback–Leibler (KL) divergence, which is generated by the negative entropy function. For sequence-form policies, we use the dilated Distance-Generating Function (dilated DGF) [21, 11, 31]:

$$d(\boldsymbol{x}_i) = \sum_{s,a \in \mathcal{S}_i \times \mathcal{A}_i} \boldsymbol{x}_i(s,a) \beta_s d_s \left( \frac{\boldsymbol{x}_i(s)}{\|\boldsymbol{x}_i(s)\|_1} \right), \tag{19}$$

where $\beta_s > 0$ is a hyper-parameter and $d_s$ is a strictly convex function defined on $\mathbb{R}^{|\mathcal{A}(s)|}$, for example, the negative entropy function. It is proven that for the dilated DGF (Lemma D.2 in [31]), we have the Bregman divergence

$$\mathcal{B}_d(\boldsymbol{x}_i \| \boldsymbol{x}_i') = \sum_{s,a \in \mathcal{S}_i \times \mathcal{A}_i} \boldsymbol{x}_i(s,a) \beta_s \mathcal{B}_{d_s} \left( \frac{\boldsymbol{x}_i(s)}{\|\boldsymbol{x}_i(s)\|_1} \bigg\| \frac{\boldsymbol{x}_i'(s)}{\|\boldsymbol{x}_i'(s)\|_1} \right). \tag{20}$$

The result allows us to compute the Bergman divergence using the behavior policy instead of explicitly using the sequence-form representation, which is prohibited in large-scale games. Formally, we define the distance of two policies $\pi_i, \pi_i'$ as

$$\text{dist}(\pi_i, \pi_i') := \mathcal{B}_d(\boldsymbol{x}_i \| \boldsymbol{x}_i') = \sum_{s,a \in \mathcal{S}_i \times \mathcal{A}_i} \left( \prod_{\widetilde{s}_i, \widetilde{a} \in \tau(s,a)} \pi_i(\widetilde{a} | \widetilde{s}_i) \right) \beta_s \mathcal{B}_{d_s}(\pi_i(s) \| \pi_i'(s)). \tag{21}$$

### B.2 Proof of Proposition 3.2

*Proof.* Recall that the PE of joint policy set $\Pi$ in two-player zero-sum games can be written into

$$\mathcal{PE}(\Pi) = \frac{1}{2} \sum_{i=1,2} \max_{\Pi_i' \subseteq \Omega_i} \mathcal{P}_i(\Pi_i', \Pi_{-i})$$

$$= \frac{1}{2} (\max_{\Pi_i' \subseteq \Omega_i} \mathcal{P}_i(\Pi_i', \Pi_2) - \min_{\Pi_{-i}' \subseteq \Omega_2} \mathcal{P}_i(\Pi_i, \Pi_2')),$$

since $\mathcal{P}_{-i}(\Pi_{-i}', \Pi_1) = -\mathcal{P}_i(\Pi_i, \Pi_2')$.

1. We can check this property directly, since

$$\mathcal{PE}(\Pi) = \frac{1}{2} (\max_{\Pi_i' \subseteq \Omega_i} \mathcal{P}_i(\Pi_i', \Pi_2) - \min_{\Pi_2' \subseteq \Omega_2} \mathcal{P}_i(\Pi_i, \Pi_2'))$$

$$\geq \frac{1}{2} (\mathcal{P}_i(\Pi_i, \Pi_2) - \mathcal{P}_i(\Pi_i, \Pi_2)) = 0$$

2. We first note that $\mathcal{H}(\Pi_1) \times \mathcal{H}(\Pi_2) \subseteq \mathcal{H}(\Pi_i') \times \mathcal{H}(\Pi_{-i}')$ means $\mathcal{H}(\Pi_1) \subseteq \mathcal{H}(\Pi_i')$ and $\mathcal{H}(\Pi_2) \subseteq \mathcal{H}(\Pi_{-i}')$.

   $\mathcal{H}(\Pi_2) \subseteq \mathcal{H}(\Pi_2')$ suggests that $\forall \Phi_i \in \Omega_i$, we have $\mathcal{P}_i(\Phi_i, \Pi_2) \geq \mathcal{P}_i(\Phi_i, \Pi_2')$, which is a property of relative population performance [2]. Take the maximum for both sides, we have,

$$\max_{\Phi_i \subseteq \Omega_i} \mathcal{P}_i(\Phi_i, \Pi_{-i}) \geq \max_{\Phi_i \subseteq \Omega_i} \mathcal{P}_i(\Phi_i, \Pi_{-i}') \tag{22}$$

   Similarly, using the condition $\mathcal{H}(\Pi_1) \subseteq \mathcal{H}(\Pi_i')$, we have,

$$\min_{\Phi_{-i} \subseteq \Omega_{-i}} \mathcal{P}_i(\Pi_i, \Phi_{-i}) \leq \min_{\Phi_{-i} \subseteq \Omega_{-i}} \mathcal{P}_i(\Pi_i', \Phi_{-i}) \tag{23}$$

   Combining Equation 22 and 23, we obtain $\mathcal{PE}(\Pi) \geq \mathcal{PE}(\Pi')$

3. By the monotonicity of PE, we have

$$\mathcal{PE}(\Pi) = \frac{1}{2}(\mathcal{P}_i(\Omega_i, \Pi_{-i}) - \mathcal{P}_i(\Pi_i, \Omega_{-i})) \tag{24}$$

Since $\Pi_{-i} = \{\pi_{-i}\}$ contains only one policy, by the definition of relative population performance, we have,

$$\mathcal{P}_i(\Omega_i, \{\pi_{-i}\}) = u(BR(\pi_{-i}), \pi_{-i}); \mathcal{P}_i(\{\pi_i\}, \Omega_{-i})) = u(\pi_i, BR(\pi_i)) \tag{25}$$

Substituting Equation 25 into Equation 24, we obtain

$$\mathcal{PE}(\Pi) = \mathcal{E}(\pi) \tag{26}$$

4. By Equation 24 and the definition of the relative population performance, there exists NE $(\sigma_i, \pi_{-i})$ in game $\mathbf{M}_{\Omega_i, \Pi_{-i}}$ and NE $(\pi_i, \sigma_{-i})$ in game $\mathbf{M}_{\Pi_i, \Omega_{-i}}$ s.t.

$$\mathcal{PE}(\Pi) = \frac{1}{2}(u(\sigma_i, \pi_{-i}) - u(\pi_i, \sigma_{-i}))$$

By the definition of NE, we have,

$$\mathcal{PE}(\Pi) = \frac{1}{2}(\max_{\pi'_i \in \Omega_i} u(\pi'_i, \pi_{-i}) - \min_{\pi'_{-i} \in \Omega_{-i}} u(\pi_i, \pi'_{-i}))$$
$$= \mathcal{E}(\pi) \tag{27}$$

We now prove $(\pi_i, \pi_{-i})$ is also the least exploitable policy in the joint policy set. $\forall \pi' = (\pi'_i, \pi'_{-i}) \in \Pi_i \times \Pi_{-i}$, we have,

$$\mathcal{E}(\pi) = \mathcal{PE}(\Pi) \leq \mathcal{PE}(\{\pi'_i\} \times \{\pi'_{-i}\}) = \mathcal{E}(\pi') \tag{28}$$

where the inequality is from the monotonicity of PE.

Hence, $(\pi_i, \pi_{-i})$ is the least exploitable policy in the Policy Hull of the population.

5. **Necessity** If $(\sigma_i^*, \sigma_{-i}^*)$ is the NE of the game, we can regard the $\{\sigma_i^*\}$ and $\{\sigma_{-i}^*\}$ as the populations which contain a single policy. Then by property 2, we have,

$$\mathcal{PE}(\Pi) \leq \mathcal{PE}(\{\sigma_i^*\} \times \{\sigma_{-i}^*\})$$
$$= \frac{1}{2}(\mathcal{P}_i(\Omega_i, \{\sigma_{-i}^*\}) - \mathcal{P}_i(\{\sigma_i^*\}, \Omega_{-i}))$$
$$= \frac{1}{2}(u(\sigma_i^*, \sigma_{-i}^*) - u(\sigma_i^*, \sigma_{-i}^*)) = 0$$

Note that by property 1, we have $\mathcal{PE}(\Pi) \geq 0$, so we have $\mathcal{PE}(\Pi) = 0$.

**Sufficiency** Using the conclusion from 3, It is easy to check the sufficiency. If $\mathcal{PE}(\Pi) = 0$, there exists $\sigma = (\sigma_i, \sigma_{-i})$ s.t. $\mathcal{E}(\sigma) = 0$, which means $\sigma$ is the NE of the game.

$\square$

### B.3 Proof of Proposition 3.3

*Proof.* Recall that at each iteration $t$, the PSRO calculates the NE $(\sigma_i^t, \sigma_{-i}^t)$ of the meta-game $\mathbf{M}_{\Pi_i^t, \Pi_{-i}^t}$, then adds the new policy $\pi^t = (BR(\sigma_{-i}^t), BR(\sigma_i^t))$ to the population.

If $PE(\Pi^t) > 0$, then by the second property in Proposition 3.2, we know the population does not contain the NE of the game. Therefore, since $(\sigma_i^t, \sigma_{-i}^t) \in \Pi^t$, it is not the NE of the full game, which means at least one of the conditions holds: (1) $BR(\sigma_{-i}^t) \notin \mathcal{H}(\Pi_i^t)$; (2) $BR(\sigma_i^t) \notin \mathcal{H}(\Pi_{-i}^t)$, which suggests the enlargement of the Policy Hull.

In finite games, since the PSRO adds pure BR at each iteration, this Policy Hull extension stops at some iteration, saying $T$, in which case we have $BR(\sigma_{-i}^T) \in \mathcal{H}(\Pi_i^T)$ and $BR(\sigma_i^T) \in \mathcal{H}(\Pi_{-i}^T)$ (by above analysis). Let $(\pi_i^{T+1}, \pi_{-i}^{T+1}) := (BR(\sigma_{-i}^T), BR(\sigma_i^T))$, we claim that $(\pi_i^{T+1}, \pi_{-i}^{T+1})$ is

the NE of the whole game. To see this, note that $(\sigma_i^T, \sigma_{-i}^T)$ is the NE of the game $\mathbf{M}_{\Pi_i^T, \Pi_{-i}^T}$ and $\pi_i^{T+1} \in \mathcal{H}(\Pi_i^T)$, hence we have,

$$u(\sigma_i^T, \sigma_{-i}^T) \geq u(\pi_i^{T+1}, \sigma_{-i}^T). \tag{29}$$

On the other hand, since $\pi_i^{T+1}$ is the BR to $\sigma_{-i}^T$, we have,

$$u(\pi_i^{T+1}, \sigma_{-i}^T) \geq u(\sigma_i^T, \sigma_{-i}^T). \tag{30}$$

Combining Equation 29 and 30, we obtain $u(\pi_i^{T+1}, \sigma_{-i}^T) = u(\sigma_i^T, \sigma_{-i}^T)$, which suggests $\sigma_i^T$ is also the BR of $\sigma_{-i}^T$. Similarly, we can prove that $\sigma_{-i}^T$ is the BR of $\sigma_i^T$, and hence $(\sigma_i^T, \sigma_{-i}^T)$ is the NE of the game.

Finally, by the fourth property in Proposition 3.2, we have $\mathcal{PE}(\Pi^T) = 0$. $\qquad\square$

### B.4 Proof of Proposition 4.3

This is an intermediate result in the proof of 3.3 (Appendix B.3).

### B.5 Proof of Theorem 3.4

*Proof.* We prove this theorem by providing the counterexamples. Let's first focus on the following two-player zero-sum game:

$$\begin{array}{c} \\ R \\ P \\ S \\ R' \\ P' \\ S' \end{array} \begin{array}{ccc} R & P & S \\ \left( \begin{array}{ccc} 0 & -1 & 1 \\ 1 & 0 & -1 \\ -1 & 1 & 0 \\ 1 & -1 & 1 \\ 1 & 1 & -1 \\ -1 & 1 & 1 \end{array} \right) \end{array} \tag{31}$$

The game is inspired by Rock-Paper-Scissors. We add high-level strategies $\{R', P', S'\}$ to player $i$ in the original R-P-S game. The high-level strategy can beat the low-level strategy with the same type.

Let $\Pi_i = \{S, R', P'\}$ and $\Pi_{-i} = \{R, P, \frac{1}{2}R + \frac{1}{2}P\}$, where $\frac{1}{2}R + \frac{1}{2}P$ means playing $[R, P]$ with probability $[\frac{1}{2}, \frac{1}{2}]$. The payoff matrix $\mathbf{M}_{\Pi_i, \Pi_{-i}}$ is

$$\begin{array}{c} \\ S \\ R' \\ P' \end{array} \begin{array}{ccc} R & P & \frac{1}{2}R + \frac{1}{2}P \\ \left( \begin{array}{ccc} -1 & 1 & 0 \\ 1 & -1 & 0 \\ 1 & 1 & 1 \end{array} \right) \end{array} \tag{32}$$

It is in player $i$'turn to choose his new strategy. Considering two candidates $\pi_i^1 = S'$ and $\pi_{-i}^2 = \frac{1}{2}R + \frac{1}{2}R'$ (There are many choices for $\pi_i^2$ to construct the counterexample, we just take one for example).

The payoff of $\pi_i^1$ against $\Pi_{-i}$ is $(-1, 1, 0)$, the same as the first row in $\mathbf{M}_{\Pi_i, \Pi_{-i}}$ The payoff of $\pi_i^2$ is $(\frac{1}{2}, -1, -\frac{1}{4})$, which is out of the gamescape of $\Pi_i$.

Denote $\Pi_i^1 = \Pi_i \cup \{\pi_i^1\}$, $\Pi_i^2 = \Pi_i \cup \{\pi_i^2\}$. By the analysis above, we have

$$\mathcal{GS}(\Pi_i^1 | \Pi_{-i}) \subsetneq \mathcal{GS}(\Pi_i^2 | \Pi_{-i}) \tag{33}$$

Hence, to promote the diversity defined on the payoff matrix and aim to enlarge the gamescape [2, 41], we should choose $\pi_i^2$ to add. However, in this case, we show adding $\pi_i^1$ is more helpful to decrease PE.

Denote $\Pi^1 = \Pi_i^1 \times \Pi_{-i}$ and $\Pi^2 = \Pi_i^2 \times \Pi_{-i}$, by Equation 24 and a few calculation, we have,

$$\mathcal{PE}(\Pi^1) - \mathcal{PE}(\Pi^2) = -\frac{1}{2}\mathcal{P}(\Pi_i^1, \Omega_{-i}) + \frac{1}{2}\mathcal{P}(\Pi_i^2, \Omega_{-i}) \approx -0.17 + 0.10 < 0, \tag{34}$$

i.e., $\mathcal{PE}(\Pi^1) < \mathcal{PE}(\Pi^2)$. We can see that adding a new policy following the guidance of enlarging gamescape may not be the best choice to decrease PE.

Back to the Proposition 3.4, since we have found an example that,

$$\mathcal{GS}(\Pi_i^1|\Pi_{-i}) \subsetneq \mathcal{GS}(\Pi_i^2|\Pi_{-i}) \quad \text{but} \quad \mathcal{PE}(\Pi^1) < \mathcal{PE}(\Pi^2) \tag{35}$$

hence,

$$\mathcal{GS}(\Pi_i^1|\Pi_{-i}) \subsetneq \mathcal{GS}(\Pi_i^2|\Pi_{-i}) \nRightarrow \mathcal{PE}(\Pi^1) \geq \mathcal{PE}(\Pi^2) \tag{36}$$

Rewrite the example in Equation 35, we have,

$$\mathcal{PE}(\Pi^2) \geq \mathcal{PE}(\Pi^1) \quad \text{but} \quad \mathcal{GS}(\Pi_i^2|\Pi_{-i}) \supsetneq \mathcal{GS}(\Pi_i^1|\Pi_{-i}) \tag{37}$$

Rename the superscripts (exchange the identity of $1, 2$ in the superscript), we obtain,

$$\mathcal{PE}(\Pi^1) \geq \mathcal{PE}(\Pi^2) \nRightarrow \mathcal{GS}(\Pi_i^1|\Pi_{-i}) \subsetneq \mathcal{GS}(\Pi_i^2|\Pi_{-i}) \tag{38}$$

Combine Equation 36 and 38, we complete the proof. $\qquad\square$

**Remark** We offer another example in a symmetric game, proving that this proposition is also valid in this type of game.

$$\begin{array}{c} \begin{array}{ccccc} A & B & C & D & E \end{array} \\ \begin{array}{c} A \\ B \\ C \\ D \\ E \end{array} \left( \begin{array}{ccccc} 0 & -1 & -0.5 & -1 & -4 \\ 1 & 0 & 0.5 & -1 & -4 \\ 0.5 & -0.5 & 0 & 0 & 4 \\ 1 & 1 & 0 & 0 & -4 \\ 4 & 4 & -4 & 4 & 0 \end{array} \right) \end{array} \tag{39}$$

Current policy set $\Pi_i = \Pi_{-i} = \{A, B\}$, the payoff is

$$\begin{array}{c} \begin{array}{cc} A & B \end{array} \\ \begin{array}{c} A \\ B \end{array} \left( \begin{array}{cc} 0 & -1 \\ 1 & 0 \end{array} \right) \end{array} \tag{40}$$

Considering $\pi_i^1 = C, \pi_i^2 = D$, the payoff of $\pi_i^1 = C$ against $\Pi_{-i}$ is $(0.5, -0.5)$, contained in $\mathcal{GS}(\Pi_i|\Pi_{-i})$, while the payoff of $\pi_i^2 = D$ against $\Pi_{-i}$ is $(1, 1)$, which is out of $\mathcal{GS}(\Pi_i|\Pi_{-i})$.

Denote $\Pi_i^1 = \Pi_i \cup \{\pi_i^1\}$, $\Pi_i^2 = \Pi_i \cup \{\pi_i^2\}$, $\Pi^1 = \Pi_i^1 \times \Pi_{-i}$ and $\Pi^2 = \Pi_i^2 \times \Pi_{-i}$, we have $\mathcal{GS}(\Pi_i^1|\Pi_{-i}) \subsetneq \mathcal{GS}(\Pi_i^2|\Pi_{-i})$. However,

$$\mathcal{PE}(\Pi^1) - \mathcal{PE}(\Pi^2) = -\frac{1}{2}\mathcal{P}_i(\Pi_i^1, \Omega_{-i}) + \frac{1}{2}\mathcal{P}_i(\Pi_i^2, \Omega_{-i}) \approx 0.17 - 2 < 0 \tag{41}$$

Hence, similar to the above analysis, we can prove the Proposition 3.4 in symmetric games.

### B.6 Proof of Theorem 4.4

*Proof.* Similar to Appendix B.3, we denote the meta-strategy at iteration $t$ as $(\sigma_i^t, \sigma_{-i}^t)$. At iteration $t$ of PSRO, if $\mathcal{PE}(\Pi^t) > 0$, then from Appendix B.3, we know that at least one of the conditions holds: (1) $BR(\sigma_{-i}^t) \notin \mathcal{H}(\Pi_i^t)$; (2) $BR(\sigma_i^t) \notin \mathcal{H}(\Pi_{-i}^t)$. Without loss of generality, we assume

$$BR(\sigma_{-i}^t) \notin \mathcal{H}(\Pi_i^t). \tag{42}$$

Then $\forall\, \hat{\sigma}_i \in \mathcal{H}(\Pi_i^t)$ and $\lambda > 0$,

$$\begin{aligned} & u(\hat{\sigma}_i, \sigma_{-i}^t) + \lambda \min_{\pi_i^k \in \mathcal{H}(\Pi_i^t)} \mathbb{E}_{s_i \sim \hat{\sigma}_i, b_2}[\mathrm{KL}(\hat{\sigma}_i(s_i)\|\pi_i^k(s_i))] \\ =& u(\hat{\sigma}_i, \sigma_{-i}^t) + 0 \\ <& u(BR(\sigma_{-i}^t), \sigma_{-i}^t) \\ <& u(BR(\sigma_{-i}^t), \sigma_{-i}^t) + \lambda \min_{\pi_i^k \in \mathcal{H}(\Pi_i^t)} \mathbb{E}_{s_i \sim \hat{\sigma}_i, b_2}[\mathrm{KL}(\hat{\sigma}_i(s_i)\|\pi_i^k(s_i))] \end{aligned} \tag{43}$$

where the first equation is because $\hat{\sigma}_i \in \mathcal{H}(\Pi_i^t)$ and the last two inequalities is due to the definition of BR and Equation 42 respectively.

**Algorithm 1** PSD-PSRO

---

**Input:** Initial policy sets $\Pi_i^1, \Pi_{-i}^1$
Compute payoff matrix $\mathbf{M}_{\Pi_i^1, \Pi_{-i}^1}$
Initialize meta policies $\sigma_i^1 \sim \mathrm{UNIFORM}(\Pi_i^1)$
**for** $t$ in $\{1, 2, ...\}$ **do**
    **for** *player* $i \in \{1, 2\}$ **do**
        Initialize $\pi_i = \pi_i^t$ and sample $K$ policies $\{\pi_i^k\}_{k=1}^K$ from Policy Hull $\Pi_i^t$
        **for** *many episodes* **do**
            Sample $\pi_{-i} \sim \sigma_{-i}^t$ and collect the trajectory $\tau$ by playing $\pi_i$ against $\pi_{-i}$
            Discount the terminal reward $u(\pi_i, \pi_{-i})$ to each state as the extrinsic reward $r_1$
            Discount $R^{KL}(\tau)$ to each state as the intrinsic reward $r_2$
            Store $(s, a, s', r)$ to the buffer, where $s'$ is the next state and $r = r_1 + r_2$
            Estimate the gradient in Equation 17 with the samples in the buffer and update $\pi_i$
        **end for**
        $\pi_i^{t+1} = \pi_i$ and $\Pi_i^{t+1} = \Pi_i^t \cup \{\pi_i^{t+1}\}$
    **end for**
    Compute missing entries in $\mathbf{M}_{\Pi_i^t, \Pi_{-i}^t}$
    Compute meta-strategies $(\sigma_i^{t+1}, \sigma_{-i}^{t+1})$ from $\mathbf{M}_{\Pi_i^t, \Pi_{-i}^t}$
**end for**
**Output:** current meta-strategy for each player.

---

Table 3: The *exploitability* $\times 100$ for populations generated by PSD with different diversity weights in Non-transitive Mixture Games. The standard error is calculated by running 5 experiments.

| $\lambda$ | 0 | 1 | 2 | 3 | 5 | 10 | 50 |
|---|---|---|---|---|---|---|---|
| EXP | $40.88 \pm 2.18$ | $2.81 \pm 0.66$ | $\mathbf{1.72 \pm 0.54}$ | $2.87 \pm 0.70$ | $3.73 \pm 0.77$ | $4.31 \pm 0.67$ | $8.22 \pm 1.73$ |

From Equation 43, we know that any policy in $\mathcal{H}(\Pi_i^t)$ will not be considered as the optimal solution in Equation 14, since they are dominated by at least one of the policy that out of $\mathcal{H}(\Pi_i^t)$, i.e., the BR. Hence, before PE reaches 0, solving Equation 14 will find a policy not in $\mathcal{H}(\Pi_i^t)$, which guarantees the enlargement of the Policy Hull. Once we can not enlarge the Policy Hull by adding the optimal solution in Equation 14, the PE of the Policy Hull must be 0, otherwise the optimal solution will be outside of the Policy Hull. $\qquad\square$

## C  Algorithm for PSD-PSRO

We provide the algorithm for PSD-PSRO in Algorithm 1, which generates the algorithm of PSRO to incorporate the estimation of the diversity in Equation 14.

## D  Additional Study

### D.1  PE on Leduc

In Figure 4, we report the *exploitability* of the meta NE on Goofspiel with 5 point cards setting.

### D.2  Ablation Study on Diversity Weight

We conduct the ablation study on the sensitivity of the diversity weight $\lambda$ on the Non-Transitive Mixture Game and Leduc Poker. In Table 3 and Table 4, we show the *exploitability* of the final population against different diversity weights in Non-Transitive Mixture Game and Leduc Poker, respectively. We can see that the $\lambda$ can significantly affect the training convergence and the final *exploitability*. A large or small $\lambda$ can lead to high *exploitability*. This suggests that properly combing the BR (implicitly enlarges the Policy Hull) and PSD (explicitly enlarges the Policy Hull) is critical to improving the efficiency at reducing the *exploitability*.

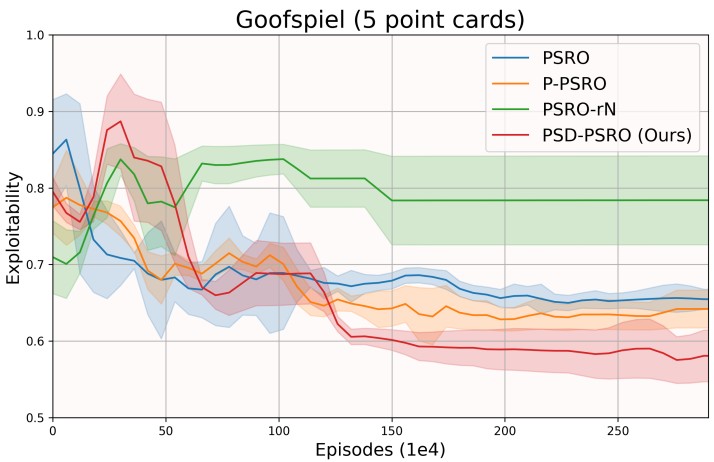

Figure 4: Exploitability on Goofspiel (5 point cards).

Table 4: The *exploitability* for populations generated by PSD with different diversity weights in Leduc Poker. The standard error is calculated by running 3 experiments.

| $\lambda$ | 0 | 0.1 | 0.5 | 1 | 2 | 5 |
|---|---|---|---|---|---|---|
| Exp | $0.590 \pm 0.017$ | $0.403 \pm 0.027$ | $\mathbf{0.356 \pm 0.012}$ | $0.398 \pm 0.030$ | $0.412 \pm 0.042$ | $0.664 \pm 0.062$ |

### D.3  The Visualization of PH

we demonstrate the expansion of PH on the Disc game [2]. In Disc game, each pure strategy is represented by a point in the circle. Due to the linearity in the payoff function, a mixed strategy is equivalent to a pure strategy and thus can be equivalently visualized. Figure 5 shows the expansion of PH on the Disc game [2]. It demonstrates that PSD-PSRO is more effective than other PSRO variants at enlarging the PH.

### D.4  Time Consumption in PSD-PSRO

We run an experiment on Leduc poker to estimate the time portion of different parts in our algorithm PSD-PSRO. We can see from Table 5 that the majority of time in PSD-PSRO is in BR solving and computing the payoff matrix, which is shared by PSRO and all its variants. The time spent on calculating our diversity metric is not significant, which accounts for $7.9\%$ computational time.

## E  Benchmark and Implementation Details

### E.1  AlphaStar888

AlphaStar888 is an empirical game generated from the process of solving Starcraft II [50], which contains a payoff table for $888$ RL policies. The occupancy measure is reduced to the action distribution as the game can be regarded as a single-state Markov Game. We choose diversity weight $\lambda$ to be $0.85$ in this game.

### E.2  Non-Transitive Mixture Game

This game consists 7 equally-distanced Gaussian humps on the 2D plane. We represent each strategy by a point on the 2D plane, which is equivalent to the weights (the likelihood of that point in each Gaussian distribution) that each player puts on the humps. The payoff of the game which contains

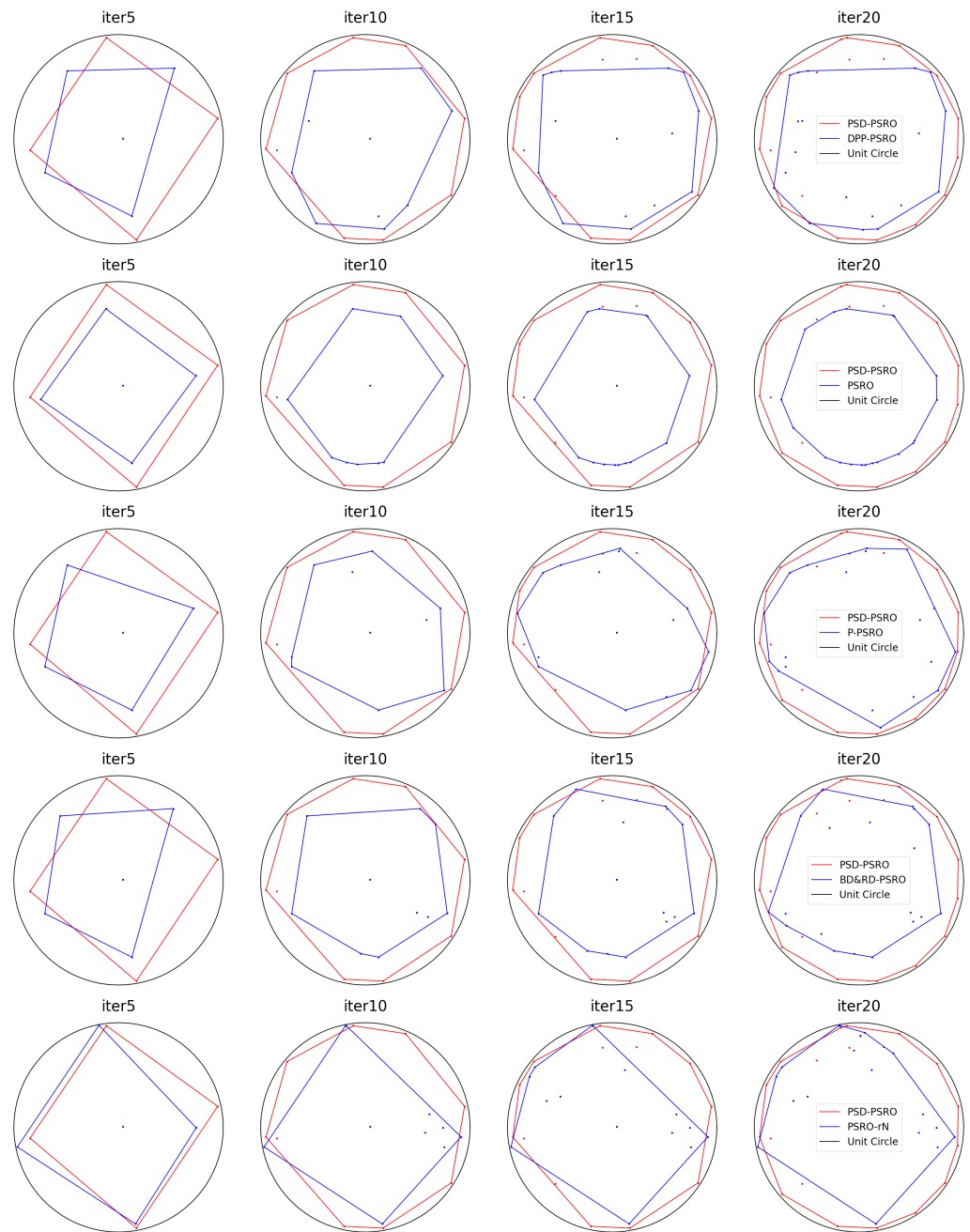

Figure 5: Equivalent visualization of PH of different methods across iterations in the Disc game

both non-transitive and transitive components is $\pi_i^T \mathcal{S} \pi_{-i} + \frac{1}{2} \sum_{k=1}^{7} (\pi_i^k - \pi_{-i}^k)$, where

$$\mathcal{S} = \begin{bmatrix} 0 & 1 & 1 & 1 & -1 & -1 & -1 \\ -1 & 0 & 1 & 1 & 1 & -1 & -1 \\ -1 & -1 & 0 & 1 & 1 & 1 & -1 \\ -1 & -1 & -1 & 0 & 1 & 1 & 1 \\ 1 & -1 & -1 & -1 & 0 & 1 & 1 \\ 1 & 1 & -1 & -1 & -1 & 0 & 1 \\ 1 & 1 & 1 & -1 & -1 & -1 & 0 \end{bmatrix}.$$

We choose diversity weight $\lambda$ to be $1.9$ in this game.

Table 5: Time Consumption in PSD-PSRO on Leduc poker

| MAIN PART OF PSD-PSRO | PERCENTAGE |
|---|---|
| COMPUTE NE OF THE META-GAME | 0.9% |
| COMPUTE THE APPROXIMATE BR | 59.9% |
| COMPUTE THE EXTRA DIVERSITY METRIC | 7.9% |
| COMPUTE THE PAYOFF MATRIX | 31.3% |

## E.3 Leduc Poker

Since the DPP-PSRO needs evolutionary updates and cannot scale to the RL setting, and the code for BD&RD-PSRO on complex games, where f-divergence on occupancy measure is approximated using prediction errors of neural networks, is not available, we compare PSD-PSRO with P-PSRO, $PSRO_{rN}$ and PSRO in this benchmark. We implement the PSRO framework with Nash solver, using PPO as the oracle agent. Hyper-parameters are shown in Table 6.

Table 6: Hyperparameters for Leduc poker

| Hyperparameters | Value |
|---|---|
| *Oracle* | |
| Oracle agent | PPO |
| Replay buffer size | $10^4$ |
| Mini-batch size | 512 |
| Optimizer | Adam |
| Learning rate | $3 \times 10^{-4}$ |
| Discount factor ($\gamma$) | 0.99 |
| Clip | 0.2 |
| Max Gradient Norm | 0.05 |
| Policy network | MLP (state_dim-256-256-256-action_dim) |
| Activation function in MLP | ReLu |
| *PSRO* | |
| Episodes for each BR training | $2 \times 10^4$ |
| meta-policy solver | Nash |
| *$PSRO_{rN}$* | |
| Episodes for each BR training | $2 \times 10^4$ |
| meta-policy solver | rectified Nash |
| *P-PSRO* | |
| Episodes for each BR training | $2 \times 10^4$ |
| meta-policy solver | Nash |
| number of threads | 3 |
| *PSD-PSRO* | |
| Episodes for each BR training | $2 \times 10^4$ |
| meta-policy solver | Nash |
| diversity weight $\lambda$ | 0.1 |

## E.4 Goofspiel

In game theory and artificial intelligence, Goofspiel is commonly used as an example of a multi-stage simultaneous move game. In two-player Goofspiel, each player has one suit of cards, which is ranked $A$ (low), 2, ..., 10, $J$, $Q$, $K$ (high). Another suit of cards serves as the prize (competition cards). Play proceeds in a series of rounds. The players make sealed bids for the top (face up) prize by selecting a card from their hand (keeping their choice secret from their opponent). Once these cards are selected, they are simultaneously revealed, and the player making the highest bid takes the competition card. If tied, the competition card is discarded. The cards used for bidding are discarded, and play continues

with a new upturned prize card. After 13 rounds, there are no remaining cards and the game ends. Typically, players earn points equal to the sum of the ranks of cards won (i.e. $A$ is worth one point, 2 is worth two points, etc., $J$ 11, $Q$ 12, and $K$ 13 points). Goofspiel demonstrates high non-transitivity. Any pure strategy in this game has a simple counter-strategy where the opponent bids one rank higher, or as low as possible against the King bid. As an example, consider the strategy of matching the upturned card value mentioned in the previous section. The final score will be 78 - 13 with the $K$ being the only lost prize. we focus on the simple versions of this game: goofspiel with 5 point cards and 8 point cards settings.

Similar to the setting in Leduc poker, we compare PSD-PSRO with P-PSRO, $PSRO_{rN}$, and PSRO in this benchmark. We implement the PSRO framework with Nash solver, using DQN as the Oracle agent. We use Linear Programming to solve NE from the payoff table. As the game size of Goofspiel is larger than Leduc poker, we improve the capacity of the model by setting the hidden dimension in MLP to be 512 and the episodes for each BR training to be $3 \times 10^4$. Other settings are similar to Leduc poker in Table 6.

