# OpenReview forum: "Policy Space Diversity for Non-Transitive Games"
_NeurIPS.cc/2023/Conference — NeurIPS 2023 poster_

### Official Review · Reviewer_Mw94 · 2023-06-16

**Soundness:** 3 good
**Presentation:** 4 excellent
**Contribution:** 2 fair
**Rating:** 7
**Confidence:** 4

**Summary:**

This paper investigates the strategy exploration problem faced in learning-based game-solving algorithms. Their posed solution is to expand the empirical game with policies that are both approximate best responses to the current solution and maximally expands the policy hull. This method is motivated by an analysis on the exploitability of the population captured by the empirical game. They show a variety of performance benefits across a suite of games.

**Strengths:**

Overall, this paper is well-rounded. The framing is clear and the related work very nicely characterizes a variety of the preceding work. While the contributed ideas of population performance and exploitability are lightly different than previous work, I think the interpretation brought forward by the small difference is rather nice. Moreover, this work treats the soft and difficult topic of diversity formally, which is a nice change of pace compared to preceding work focused on heuristics. They provide both an exact formulation of their metric, and a practical version that can be used in gradient optimization.

**Weaknesses:**

The major weakness of this work is the confusion of empirical games and what they refer to as the "meta game". The crucial difference being that empirical games have only estimates of the true payoffs. PSRO uses an empirical game, but meta games may be useful for best-case convergence guarantees. Unfortunately this confuses and invalidates a variety of claims throughout the paper that mix up their usage. This is easy to correct, and I am very supportive of the paper post this correction.

Another general point of weakness is the recurring claim of the problems with gamescapes. They are right, as they showed, that each iteration isn't gauranteed to lead to a monotonically improving game solution. However, this does not mean that the gamescape analysis wouldn't have long-term benefits. I suggest lightening the critique of gamescapes, and instead focus criticism only on their specific application of per-iteration policy exploration.

I would have liked to see some analysis related to diversity and/or settings of their algorithm in the results.

**Questions:**

- Convex Hull Enlargement and Occupancy Measure Mismatching appear to be functionally different from the other diversity measures because they focus on the impact of including another policy. I was wondering if you thought about how your methods or the others might be alternatively used between these two perspectives?
- Population exploitability appears to be similar to the metrics defined in Lanctot et al.. "Population-based Evaluation in Repeated Rock-Paper-Scissors as a Benchmark for Multiagent Reinforcement Learning." Might be worth discussing as concurrent/previous work.
- L159-160: There's nothing ill-posed with the idea of enlarging a gamescape generally. I would recommend refining this claim to focus on your point: for diversity.
- L165-166: Exploitability increasing in the short-term might be OK if it facilitates better long-term exploration. I would suggest revising this claim to focus on when future iterations occurring are uncertain---I believe AnytimePSRO makes an argument like this.
- L197: If you're defining the concept of fine policy representation could you make it more clear please?
- L218: There is something potentially confusing about your definition of the sequence-form. Shouldn't it either include all possible trajectories to (s,a) or be only the probability chain of actions since perfect recall makes all paths unique? A quick comment about this in the paper I think would be very nice.
- L245: This method requires a design choice of which opponent distribution diversity will be measured against. Could the authors comment on different settings they tried? It might be also useful to discuss how this is both a limit and potential advantage of this method when compared to other diversity metrics. You are both potentially more limited in your measure with a bad choice of opponent, or able to better measure by including behaviors outside of the empirical game.
- For the AlphaStar888 experiments could you explain how you computed the diversity metrics given only the empirical game? The metrics are all dependent on the policies that are not directly available.
- L323: I would be interested in having the details of what makes BD&RD-PSRO to tune and anything you might have learned from the attempted reimplementation in the appendix.
- For Goofspiel, if you can easily get them, could you please also provide approximate regret curves? This can be estimated using the combined empirical game across methods.
- For Goofspiel, are you using the final solutions produced by the algorithms to compare or regret? It's also not clear how long this experiment has been run and the respective population sizes.


**Limitations:**

There is no focused discussion on either the limitations or broader impact of this work within the paper.

The last paragraph, presumably their limitation sections, quickly discuss the role of trading-off diversity in game solving. I would have preferred to see discussions related:
 - Setting the opponent policy for policy comparisons
 - How the selection of the aforementioned opponent biases the measure of diversity
 - How changes in the opponent policy will require much larger computational costs
 - Results are partially on games that are not exactly solved, and the true relative performance of methods may differ
 - etc..

---

> ### Author Rebuttal · Authors · 2023-08-08
>
> Thanks very much for your review. We respond to all your questions below.
>
> **Q1.** the confusion of empirical games and the "meta game".
>
> **A1.**  The "meta game" will be used more cautiously in the revision. The main purpose of our introduction of "meta game" was to define the meta-policy (a discrete probability distribution over policies).
>
> **Q2.** the recurring claim of the problems with gamescapes.
>
> **A2.** We agree that the critique of gamescapes appears in an excess of
> places in the paper (we will lighten this). Yet, what we really want to emphasize is enlarging the gamescape does not necessarily improve the approximation to a NE either in one iteration on in the long run (Theorem 3.4). By contrast, optimizing our diversity (policy space diversity) defined on the policy hull (instead of gamescape) improves the approximation to a NE (Theorem 4.2).
>
> **Q3.** Convex Hull Enlargement and Occupancy Measure Mismatching. How your methods relate to them?
>
> **A3.** There is the subtle difference between the definition of a diversity metric and the practical way of optimizing the diversity metric.
> We would like to argue the superiority of both our definition of diversity metric and our practical way to optimize our diversity metric, in comparison to all previous diversity-enhancing PSRO methods.
>
> - Our diversity metric is defined on the policy hull, the optimization of which has theoretical and positive impact on approaching a NE (Theorem 4.2). In contrast, all previous diversity-enhancing PSROs defined their diversity metrics on the gamescape, the optimization of which does not necessarily mean a better approximation to a NE (Theorem 3.4).
>
> - To quantify the similarity of two policies, a number of previous methods focus on the f-divergence of the occupancy measure [19,26]. However, computing the f-divergence based on the occupancy measure is usually intractable and often in practice **roughly** approximated using neural networks [19,26]. By contrast, we employ the sequence form representation of a policy, which is widely used in the study of imperfect-information games. More importantly, we deduce a practical **state-based** diversity optimization objective (Eq 14) based on the Bregman divergence on the sequence form representation.
>
> **Q4.** Population exploitability appears to be similar to the metrics defined in Lanctot et al..
>
> **A4.** We will add the discussion to the revision.
>
> **Q5.** There's nothing ill-posed with the idea of enlarging a gamescape generally. I would recommend refining this claim to focus on your point: for diversity.
>
> **A5.** We agree that there is nothing ill-posed if our goal is solely about enlarging a gamescape. What we would like to emphasize is enlarging the gamescape does not necessarily lead to a better approximation to a NE (theorem 3.4) (the goal of most previous diversity-enhancing PSRO variants). PSD-PSRO improves PSRO with our policy space diversity, and its only goal is to find a NE (not for diversity).  We will make this clearer in the revision.
>
> **Q6.** Exploitability increasing in the short-term might be OK if it facilitates better long-term exploration.
>
> **A6.** We would like to clarify that PSD-PSRO is more about a new diversity regularization that has proved benefits in approximating a NE, either in one iteration or the long term. In contrast, previous diversity metrics (mainly on gamescapes) for PSRO do not have the positive&theoretical connection to the goal of approximating a NE.  BTW, we believe Anytime PSRO is targeting reducing the exploitability at each iteration. We will make this clearer in the revision.
>
> **Q7.** If you're defining the concept of fine policy representation could you make it more clear please?
>
> **A7.** Yes, we will.
>
> **Q8.** There is something potentially confusing about your definition of the sequence-form.
>
> **A8.**  Yes, perfect recall makes all paths unique. We will make it clearer.
>
> **Q9.** This method requires a design choice of which opponent distribution diversity will be measured against.
>
> **A9.** Actually, the sequence form definition (Eq 11) does not depend on the opponent policy while the occupancy measure does (because it is a prob distribution over state-action). According to Eq 14, we do need a opponent policy $b_{-i}$ to generate the trajectories, and occupancy measure mismatch or diversity on gamescape need opponent policies as well.  In all our experiments, we set $b_{-i}$ to $\sigma_{-i}^{t}$ (the meta NE policy from last iteration) for convenience. In the rebuttal pdf, we add ablation experiments on Leduc setting $b_{-i}$ to the uniform random policy, where PSD-PSRO still plays the best.
>
> **Q10.** Question about AlphaStar888.
>
> **A10.** AlphaStar888 can be viewed as a zero-sum symmetric two-player game (like the rock-paper-scissors), where there is only one state $s_0$. In $s_0$, there are 888 legal actions. Any mixed strategy is a discrete probability distribution over the 888 actions. So, the diversity term in Eq 14 for AlphaStar888 reduces to the KL divergence between two 888-dim discrete probability distributions.
>
> **Q11.**  I would be interested in having the details of what makes BD&RD-PSRO to tune
>
> **A11.** The code is not available according to https://github.com/sjtu-marl/bd_rd_psro, where it says "It is the company's internal code, which cannot be disclosed here due to the copyright issue.". We have not tried reimplementing BD&RD-PSRO when f-divergence is approximated using prediction error of neural network (this is the reason we feel it is hard to tune).
>
> **Q12.** Question about Goofspiel.
>
> **A12.** The results on Goofspiel (8 point cards, too large to show exact exploitability in OpenSpiel) in the main paper are the comparison among final solutions produced by different methods. We have added in the rebuttal pdf the results on Goofspiel (5 point cards) with the exact convergence of exploitability of different methods. More experiment details will be added in the revision.

---

> > ### Comment · Reviewer_Mw94 · 2023-08-11
> >
> >
> > Thank you for addressing some of my questions. I would like to follow-up on several of the questions that I feel are still unresolved:
> > - Q3: In the question about Convex Hull Enlargement and Occupancy Measure Mismatching I did not mean to request you defend the novelty or superiority of proposed methods. My question is specifically that these metrics "focus on the impact of including another policy" as opposed to measuring diversity of say a fixed set or between two policies. I was curious if you had any _thoughts_ or _speculations_ about the role this might play in their performance? Moreover, have you _thought_ about how other diversity metrics might be recast to focus on "the impact of including another policy", or how your methods might be recast in other perspectives?
> > - Q6: I agree with the authors intention, which is to say that adding a single policy can reduce performance. However, the point I'm making ontop of that is this not necessarily a bad thing. If you know you're going to continue the game solving process and add more policies in the future, taking a loss in the short term for larger gains in the long term (exploration v exploitation) could be very advantageous. I am suggesting that the discussion in L164-166 consider this instead of implying that this is a strictly bad result. Does this make sense?
> > - Q10: Could the authors link where the acquired the AlphaStar888 data? I've tried to track it down and cannot easily find it. It looks like the github you linked might have the data, but how did they end up with it? They're different authors from the StarCraft team. It would be nice to have traceability of this data and confirmation that this is the version of the data you're using. I urge the authors to cite that this is the source of the data and mention it explicitly in their work.
> > - Q11: If you did not attempt at all to replicate their work then I would modify L322-323 in the paper, because I believe its phrased in a misleading way. "BD&RD-PSRO is hard to tune (the code is not available)" to me suggested that you had _tried_ to tune the algorithm. I would have preferred if the authors had attempted to replicate it to the best of their ability, as it is a very pertinent baseline, and then provided discussion about any difficulties associated with it in the text. As of now the claim in the paper "is hard to tune" is strictly speculation---and frankly, most deep reinforcement learning methods are hard to tune, so I don't think this is particularly unique. I realize at this point in the process it's unreasonable to run this experiment, so at least the language in the paper needs to be corrected. I would also strongly urge the authors to actually attempt to run this baseline, especially for future versions of this paper.

---

> > > ### Author Response · Authors · 2023-08-12
> > > **Many thanks for your reply**
> > >
> > > We provide responses to your remaining questions. We would be happy to discuss if there is still anything unclear.
> > >
> > > **A3**: Thanks for your clarification. Within the context of PSRO, where new policies are iteratively added to the population, we believe it is more straightforward to have a diversity (for the purpose of NE solving) that evaluates the impact of a new added policy, as opposed to any other form of diversity (e.g., measuring diversity of a fixed set or between two policies). Yet, these are all connected in the sense that one form of diversity can be written in another form. For instance, the impact (with respect to a population $\Pi$) of policy $\pi_a$ versus $\pi_b$  can be written as Div($\Pi \cup \pi_a$) versus Div($\Pi \cup \pi_b$).
> > >
> > > In PSRO, how to evaluate the **impact** of introducing a new policy (say $\pi_i$)  plays an **important** role for the final performance (finding a full game NE). There are two aspects in our argument:
> > >
> > > - There should be an anchor (either a single policy or a population of policy), when we talk about the impact of $\pi_i$. We think for PSRO the anchor is better to be a population than any fixed single policy, simply because PSRO is a population-based algorithm (the final output solution is produced by the entire population, i.e., the meta NE). Nonetheless, it is worth mentioning that the impact of $\pi_i$ with respect to a population $\Pi^{t}_i$ is often approximated by a min policy $\pi^{min}_i$ 'contained' in $\Pi^{t}_i$: distance($\pi_i$, $\pi^{min}_i$).
> > >
> > > - When we talk about the similarity between two policies, there should be a metric. First, we need a representation of policy. For the purpose of approximating a NE, we define what is *a fine policy representation* (Line 205), which should have the property of linearity and bijection. For instance, the pay-off vector in the gamescape is **not** *a fine policy representation*, while the sequence form (Eq 11) and the occupancy measure are.  Afterwards, we need a similarity function on the representation. Previous work on occupancy measure uses the f-divergence, while we use the Bregman divergence.
> > >
> > >
> > > Have you thought about how other diversity metrics might be recast to focus on "the impact of including another policy".
> > >
> > > - The occupancy measure mismatch between $\pi_i$ and $\Pi^t_i$ might be recasted for our purpose. However, as far as we know in the current literature, f-divergence on occupancy measure is roughly approximated using predictions of neural networks. In contrast, we have a closed form for our Bregman divergence on the sequence form (Eq 13).
> > >
> > > how your methods might be recast in other perspectives?
> > >
> > > - We believe our definition of *fine policy representation* might be helpful for other purposes as well, because it provides a guideline on how to represent a policy properly.
> > > - We believe our Bregman divergence on the sequence form (Eq 13) may be useful for clustering policies, which can be helpful to downstream tasks, such as opponent modeling.
> > >
> > >
> > > **A6**: It makes absolute sense that "taking a loss in the short term for larger gains in the long term". The problem with diversity on gamescape is enlarging the gamescape in either short term or long term does not **necessarily** lead to a better approximation to a NE, . We understand that the current writing in L164-166 makes you think we are implying that taking a loss in the short term is a strictly bad thing. Thanks for pointing this out, and how about we change L164-166 to the following: "In other words, enlarging the gamescape in either short term or long term does not necessarily lead to a better approximation to a full game NE."
> > >
> > >
> > > **A10**: We acquired the AlphaStar888 data from https://github.com/oslumbers/diverse-psro. We have sent an email to ask one of the authors of diverse PSRO for the exact source (presumably from DeepMind) of data. Many thanks for your suggestion, and we will cite the DeepMind source of data in the revision once we get it.
> > >
> > > **A11**: Thanks for your suggestion. We would change the text "BD&RD-PSRO is hard to tune" to "the code for BD&RD-PSRO on complex games, where f-divergence on occupancy measure is approximated using prediction errors of neural networks, is not available". BTW, we did compare to BD&RD-PSRO on simpler games in Figure 1&2 in the paper. Reimplementation of the full version of BD&RD-PSRO is definitely one of our future work.

---

> > > > ### Comment · Reviewer_Mw94 · 2023-08-14
> > > >
> > > > Thank you for answering my questions and entertaining my request for speculation. I think these changes will help future readers of the paper.
> > > >
> > > > I am happy to move my score up to a 7.

---

> > > > > ### Author Response · Authors · 2023-08-14
> > > > >
> > > > > It is our pleasure, and many thanks for your review and constructive suggestions.

---

> > > > > ### Author Response · Authors · 2023-08-16
> > > > >
> > > > > Dear Reviewer Mw94, could you please update your change of score ($6 \rightarrow 7$) in your Official Review at your convenience?
> > > > > Many thanks again for your time and constructive suggestions. We really appreciate it.

---

### Official Review · Reviewer_YW2F · 2023-06-28

**Soundness:** 1 poor
**Presentation:** 3 good
**Contribution:** 2 fair
**Rating:** 5
**Confidence:** 4

**Summary:**

This paper considers PSRO framework for approximating a NE in multi-agent non-stransitive games. The authors points out that the existing diversity matrics cannot guarantee that a more diverse policy space leads to a better approximation of the NE. To solve this issue, this paper proposes a new diversity metric and a corresponding algorithm, PSD-PSRO, which enjoys better convergence property.


**Strengths:**

The paper is very well-organized and easy to follow. It provides a nice overview of related literature and states its contribution clearly.


**Weaknesses:**

My major concern is the correctness of the result. In specific, I think the derivation of the gradient caculation in equation (15) might be incorrect. The derivation from line 484 to 485 in the supplementary material might need some reconsideration. I particular I don't think the first equation actually holds because the $\pi_i^{\min}$ should be a function of $\pi_i$. It might even be questionable whether this function is differentiable.

I'm also curious about the computational complexity of the algorithm proposed in the paper. It seems that compared with other diversity metrics, the new metrics proposed is harder to compute because it needs to solve a minimization problem. What are the computational time for the numerical examples? And how does it compare with other methods? In Section D.3, the authors only provides the percentage, but it would be better to have a comparison among different methods.


**Questions:**

See the Weakness section

**Limitations:**

See the Weakness section

---

> ### Author Rebuttal · Authors · 2023-08-08
>
> Thanks very much for your review. We respond to all your comments of weaknesses in the following and will update the paper to make it clearer.
>
> **Q1** I think the derivation of the gradient calculation in equation (15) might be incorrect.
>
> **A1** We double-checked the derivation of equation 15 (in appendix Line 484-485, Eq 21), and we believe it is **correct** (though we will edit to make this derivation clearer). $\pi_i^{min}$ corresponds to the nearest policy in the policy hull of  $\mathcal{\Pi^t_i}$ to $\pi_i$ (as defined in line239). $\pi_i^{min}$ indeed depends on $\pi_i$. Yet, given the current $\pi_i$, $\pi_i^{min}$ is fixed (suppose it is unique) and can be approximated **in advance**. More specifically for Eq 21:
>
> 1. In L484, since $\min_{\pi_i^k\in H(\Pi_i^t)} E_{s_i \in \tau, \tau \sim \pi_i, b_{-i}} [KL(\pi_i(s_i) \| \pi_{i}^{k}(s_i))]= E_{s_i \in \tau, \tau \sim \pi_i, b_{-i}} [KL(\pi_i(s_i) \| \pi_{i}^{min}(s_i))]$, we can get the first line.
> 2. The second line is by the definition of expectation, and the third line is the application of the product rule.
> 3. We use the policy gradient theorem for the left term in the last second line, and rewrite the right term using the notation $ E_{s_i \in \tau, \tau \sim \pi_i, b_{-i}} $.
> 4. The last term in Eq 21, which is a gradient with respect to the parameters of $\pi_i$ of a KL between $\pi_i$ and $\pi^{min}_i$. Here, the action probabilities of $\pi^{min}_i$ in $s_i$ have been evaluated in advance and are constants (not a function of $\pi_i$). As a result, this term is a function of only $\pi_i$. Also, to get the gradient properly,  a small prob (1e-5) is added to each action (normalizing afterwards) to ensure that there is no zero-prob action in both $\pi_i$ and $\pi^{min}_i$. This new detail will be added to the revision.
>
> We would be happy to provide further details if anything remains unclear.
>
> **Q2** I'm also curious about the computational complexity of the algorithm proposed in the paper.  And how does it compare with other methods? In Section D.3, the authors only provides the percentage, but it would be better to have a comparison among different methods.
>
> **A2**   As we show in Appendix D.3,  the majority time in PSD-PSRO is on BR solving and computing the payoff matrix, which is **shared** by PSRO and all its variants. The time spent on calculating our diversity metric is not significant (accounts for 7.9\% computational time). We will make this clearer in the revision.
>
>
> **Q3** It seems that compared with other diversity metrics, the new metrics proposed is harder to compute because it needs to solve a minimization problem.
>
> **A3** Ideally, to get the true $\pi_i^{min}$, we should solve a convex optimization problem. In practice, we sampled multiple policies in the policy hull of $\mathcal{\Pi_i}$ and used the sampled min policy to approximate the true $\pi_i^{min}$. This is straightforward to implement and does not add too much complexity. We will make this clearer in the revision.

---

> > ### Comment · Reviewer_YW2F · 2023-08-13
> >
> > Thank you for your response! I think A2 and A3 have addressed my questions with respect to the computational complexity.
> >
> > However, I am still not convinced by the argument in A1. In specific I don't think step 4 in A1 is convincing enough for me. I guess my question is still, since the gradient is taken on $\pi_i$ and that $\pi_i^{min}$ is a function on $\pi_i$, how come the gradient term can ignore the gradient on $\pi_i^{min}$? Also the response says ''$\pi_i^{min}$ indeed depends on $\pi_i$" yet also mentioned in step 4 that the term is "not a function of $\pi_i$". I'm quite confused by the seemingly conflicting arguments. It would be nice if the authors could write down some more rigorous arguments to show that the proof is correct.

---

> > > ### Author Response · Authors · 2023-08-14
> > > **The fourth point in A1**
> > >
> > > Thanks very much for your reply.
> > >
> > > The last term in Eq 21 ($\mathbb{E}[\nabla \text{KL}(\pi_i(s_i) \| \pi_{i}^{min}(s_i))]$), which is a gradient with respect to the parameters of $\pi_i$ of a KL between $\pi_i$ and $\pi_i^{min}$. As we said in **A3**, we  sampled multiple policies in the policy hull of $\Pi_i^t$ and used the sampled min policy (say $\hat{\pi}_i^{min}$) to approximate the true $\pi_i^{min}$:
> > > - We first obtain the fixed policy $\hat{\pi}_i^{min}$.
> > > - Then, the last term in Eq 21 becomes $\mathbb{E}[\nabla \text{KL}(\pi_i(s_i) \| \hat{\pi}_{i}^{min}(s_i))]$, where $\hat{\pi}_i^{min}$ is a **constant**.
> > >
> > > More precisely, if we undstand it correctly, the reviewer's question is whether the above two steps make sense in approximating $ \nabla \min_{\pi_i^k} \mathbb{E} [ \text{KL}(\pi_i(s_i) \| \pi_{i}^{k}(s_i))]$. The answer is yes:
> > >
> > > - In Equation 15 in the paper, we have used a property that $\nabla_{\theta} \mathbb{E}[\text{KL}(\pi_i(s_i; \theta) \| \pi_i^{min}(s_i))] \in \partial_{\theta} \min_{\pi_i^k} \mathbb{E}[\text{KL}(\pi_i(s_i; \theta) \| \pi_{i}^{k}(s_i))]   $, where $\pi_i^{min}(s_i) = \arg\min_{\pi_i^k} \mathbb{E}[\text{KL}(\pi_i(s_i; \theta) \| \pi_{i}^{k}(s_i))]  $.
> > > - To show the correctness of the equation above, we can prove a proposition as follows:
> > > > *Proposition*: For any locally Lipschitz continue function $f(x,y)$, assume $\min_y f(x, y)$ is well-defined, then
> > > $ \partial_x f(x,y) |\_{y \in \arg\min f(x, y)} \in  \partial_x \min_{y} f(x, y)$ (1),
> > > where $\partial f$ is the generalized gradient [1], which is the convex hull of the limits of the form $\lim \nabla f (x + h_i) $ where $h_i \rightarrow 0$.
> > > - The proposition is an immediate result according to [1].
> > > > *proof*:  According to Theorem 2.1 (property (4)) in [1], (1) is immediate, as $\partial_x \max_{y} g(x, y)$ (or $\partial_x \min_{y} f(x, y)$) is the convex hull of $\\{\partial_x g(x,y) |{y \in \arg\max g(x, y)} \\}$.
> > > - Note that the generalized gradient $\partial_x f(x, y)$ is the set of subgradients when $f(x, y)$ is convex on $x$ [1] (Proposition 1.2).
> > > - This theory has been used widely in existing literature. For example:
> > > 1. GAN (Proposition 2) [2], which shows that the subgradient of the generator G can be obtained by computing the gradient of G at the optimal discriminator D.
> > > 2. ED (Theorem 1) [3], which proves that the gradient of $\pi_{\theta_i}$'s value against a best response, $\beta = b_{-i}(\pi_{\theta_i})$, is a generalized gradient of $\pi_{\theta_i}$'s worst-case value function, i.e.,
> > > $\nabla_{\theta_i} v_{i}(\pi_{\theta_i}, \beta)|\_{\beta = b_{-i}(\pi_{\theta_i})} \in \partial_{\theta_i} \min_{\pi_{-i}} v_{i}(\pi_{\theta_i}, \pi_{-i})$
> > >
> > > Hope the above clarifies your concern, and we will make it clearer in the revision. We would be happy to discuss if there is still anything unclear.
> > >
> > >
> > > [1] Clarke, Frank H. "Generalized gradients and applications." Transactions of the American Mathematical Society 205 (1975): 247-262.
> > >
> > > [2] Goodfellow, Ian, et al. "Generative adversarial nets." Advances in neural information processing systems 27 (2014).
> > >
> > > [3] Lockhart, Edward, et al. "Computing approximate equilibria in sequential adversarial games by exploitability descent." arXiv preprint arXiv:1903.05614 (2019).

---

> > > > ### Comment · Reviewer_YW2F · 2023-08-14
> > > >
> > > > I sincerely thank the authors for the detailed explanation! I think now I am convinced that the proofs as far as I read make sense to me. It would be nice if the authors could add the elaboration (especially add the proposition from Theorem 2.1 in [1]), which might help readers like me to better understand how the derivation is done. I'm changing my evaluation score to 5 based on the conversation.

---

> > > > > ### Author Response · Authors · 2023-08-15
> > > > >
> > > > > It is our pleasure, and many thanks for your insightful review. We will refine the paper according to your suggestions.

---

### Official Review · Reviewer_jhEH · 2023-07-03

**Soundness:** 3 good
**Presentation:** 3 good
**Contribution:** 2 fair
**Rating:** 5
**Confidence:** 3

**Summary:**

In this work, the authors sought to justify the benefit of diversity by investigating properties of an expanding policy hull (PH), as supported by a set of policies in the context of population learning using Policy Space Response Oracle (PSRO). The authors discussed the limitations of expanding gamescape as a proxy for exploration in the policy space and reviewed several works on inducing diversity in prior works. Finally, the authors proposed their own diversity regularisation techniques and showed evidence that this approach leads to faster convergence to full game NE compared to baselines.

**Strengths:**

* originality / significance
I find the discussion on what to focus on when it comes to defining policy space diversity interesting, thorough and well motivated. There have been several works on this topics in the literature (as surveyed by the authors) but prior works presented different definitions and motivations for the need for policy space diversity which this work attempts to address. Having a unified language to discuss diversity with the goal of game-solving is sorely needed by the community.

* quality / clarity
Empirical results of PSD-PSRO appear thorough, though I have a few clarifying questions on the results. I would encourage the authors to revise the presentation to make the experimental protocol more clear and convincing. The theoretical justification for policy hull expansion appears sound.

**Weaknesses:**

* Clarity: the authors presented prior works from the angle of NE game-solving. However I believe this is not necessarily the stated goal in all prior works? For instance, PSRO_rN was proposed as a way to expand gamescape (rather than a method that solves for a full game NE at faster convergence rate). This should be clear from their discussion on the disc game where focusing on NE game solving would lead to a contracted gamescape, i.e. the final solution would not contain pure strategies that would be interesting to represent as they optimally exploit certain opponents (as compared to the NE mixed strategy). I would encourage the authors to revisit the stated motivation for these prior works and re-state their motivations more precisely.

* Scope: the scope of this work could be stated more precisely? Several prior works on population diversity explicitly focused on symmetric two-player zero-sum games (i.e. gamescape was defined in this regime). The authors appear to state their results generally (e.g. L103, $\Pi_i$, $\Pi_{-i}$), do you aim to make statements about n-player general-sum games in this paper? If so, how should one interpret e.g. Equation 7? Should I interpret $\sigma_{-i}$ as the joint distribution across all coplayers of player $i$? What would be the dimensions of the relative population performance in that case?

* Preciseness: in several places the authors seem to imply that vanilla PSRO would not converge to a full game NE while there are convergence guarantees when using suitable best-response solvers. Perhaps revisit these statements to make it clear that the intention is to offer faster convergence rate? E.g. L266 and L174.

**Questions:**

* Eq 14: how is the min operator implemented as there are infinite such mixed policies? If each such sample corresponds to a mixture-policy mixed at the start of an episode, would you draw multiple samples from the same distribution in the population simplex?

* Policy Hull: is the definition of Policy Hull the same as Population Simplex described in https://arxiv.org/pdf/2205.15879.pdf? I.e. they both correspond to all convex mixtures supported by a set of policies? If so, would simplex-NeuPL be a useful representation scheme when searching for diverse BR policies? If not, could you clarify their differences?

* L269: ... PSD-PSRO over PSRO in approximating a NE. Is this statement empirical or do you expect better convergence rate in theory which can be stated more explicitly?

* Fig 1 (Left): nit: could you specify finer-grained y-axis for these figure?

* L302: instances of Goofspiel as implemented in OpenSpiel could be solved exactly, could the authors clarify which game setting is used? Could similar analysis be done in more restricted setting such that we can show exact convergence in Goofspiel?

* AlphaStar888: the dataset presents the payoff between 888 RL agents, but the diversity regularisation term seem to suggest needing access to state action in the underlying game. Could you clarify how diversities between these RL policies are computed?

* Fig 3: exploitability in Leduc Poker seems to be reducing still (at 0.2 currently), would it reduce further to lower level? Are there computational bottleneck that would prevent further training? How are best-responses solved in this experiment? By RL? Or by exact solvers from OpenSpiel?

My reservation in accepting this work is primarily due to my confusion as stated above, I would encourage the authors to further clarify their empirical results and experimental settings.

**Limitations:**

This work is methodological and does not present immediate potential for negative societal impact.

---

> ### Author Rebuttal · Authors · 2023-08-08
>
> Thanks very much for your review. We respond to all your comments and questions below.
>
> **Q1.** The weakness of Clarity. I would encourage the authors to revisit the stated motivation for these prior works and re-state their motivations more precisely.
>
> **A1.** Many thanks for your suggestion. We will update the paper and state the goal of PSRO_rN more precisely. Yes, other than the NE game-solving, PSRO_rN is also explicitly targeting the policy population diversity defined on the gamescape.
>
> **Q2.** The weakness of Scope. the scope of this work could be stated more precisely?
>
> **A2.** Many thanks for your suggestion. The scope of this work can be stated more precisely: we focus on two-player zero-sum games. We did not mean to state our results beyond two-player zero-sum games.
>
> **Q3.** The weakness of Preciseness.  In several places the authors seem to imply that vanilla PSRO would not converge to a full game NE while there are convergence guarantees when using suitable best-response solvers.
>
> **A3.** Many thanks for your suggestion. The intention of our method PSD-PSRO is indeed to **accelerate** PSRO in finding a NE. We did not mean to imply that PSRO would not converge. In fact,  PSRO would find a full game NE once it converges. By contrast, it is not clear, in other state-of-the-art diversity-enhancing PSRO variants [1, 35, 26, 27] (citation index in the paper), whether a full game NE is found once they are converged in terms of their optimization objectives. Notably, as pointed out in [30] (citation index in the paper), PSRO_rN is not guaranteed to find a NE once converged. Again, we are aware that PSRO_rN is not solely about NE game-solving.
>
> **Q4.** Eq 14: how is the min operator implemented as there are infinite such mixed policies? would you draw multiple samples from the same distribution in the population simplex?
>
> **A4.** In practice, we sampled multiple policies (the exact number equals to the size of $\Pi^t_i$) in the policy hull of $\Pi^t_i$ and used the sampled min policy to approximate the true $\pi_i^{min}$.  Each such sample corresponds to a mixture-policy mixed at the start of an episode. We only need to **evaluate** (no need of play) the sampled mixture-policy on trajectories generated by playing $\pi_i$ against a fixed opponent $b_{-i}$ (see Eq 14). We will make this clearer in the revision.
>
>
> **Q5.** Policy Hull: is the definition of Policy Hull the same as Population Simplex described in https://arxiv.org/pdf/2205.15879.pdf?
>
> **A5.** The definition of Policy Hull is the same as Population Simplex described in https://arxiv.org/pdf/2205.15879.pdf
> The neural architecture of simplex-NeuPL is orthogonal to the contributions of our method PSD-PSRO. It is definitely worth trying this architecture in PSD-PSRO. Also, we believe the diversity optimization scheme in PSD-PSRO could be helpful to simplex-NeuPL for its own purpose as well. We will add the discussion in the revised paper.
>
> **Q6.** L269: ... PSD-PSRO over PSRO in approximating a NE. Is this statement empirical or do you expect better convergence rate in theory which can be stated more explicitly?
>
> **A6.** This statement is empirical. A better convergence rate in theory of PSD-PSRO (or other algorithms) over PSRO is an importnat future work.
>
> **Q7.** Fig 1 (Left): nit: could you specify finer-grained y-axis for these figure?
>
> **A7.** Yes. This will be made clearer in the revision.
>
> **Q8.**  could the authors clarify which game setting is used about Goofspiel? Could similar analysis be done in more restricted setting such that we can show exact convergence in Goofspiel?
>
> **A8.** Yes, the goofspiel results we have provided in the main paper is the goofspiel with 8 point cards, corresponding to the setting "turn_based_simultaneous_game(game=goofspiel(imp_info=True,num_cards=8,points_order=descending))" in open-spiel.
> The goofspiel results with exact exploitability convergence (please check the rebuttal pdf at the top) we add in the rebuttal is the goofspiel with 5 point cards, corresponding to the setting "turn_based_simultaneous_game(game=goofspiel(imp_info=True,num_cards=5,points_order=descending))" in open-spiel.
>
> **Q9.** AlphaStar888: the dataset presents the payoff between 888 RL agents, but the diversity regularisation term seem to suggest needing access to state action in the underlying game. Could you clarify how diversities between these RL policies are computed?
>
> **A9.** AlphaStar888 can be viewed as a zero-sum symmetric two-player game (like the rock-paper-scissors), where there is only one state $s_0$. In $s_0$, there are 888 legal actions. Any mixed strategy is a discrete probability distribution over the 888 actions. So, the diversity term in Eq 14 for AlphaStar888 reduces to the KL divergence between two 888-dim discrete probability distributions. This will be made clearer in the revision.
>
>
> **Q10.** Fig 3: exploitability in Leduc Poker seems to be reducing still (at 0.2 currently), would it reduce further to lower level? Are there computational bottleneck that would prevent further training? How are best-responses solved in this experiment? By RL? Or by exact solvers from OpenSpiel?
>
> **A10.** There is no computational bottleneck that would prevent further training. Longer training curves are provided in the rebuttal pdf at the top.  Best-responses for Leduc are exact solvers from OpenSpiel.
>
> Hope the above clarifies your confusion. If there is anything still unclear, please do not hesitate to bring it up.

---

> > ### Comment · Reviewer_jhEH · 2023-08-11
> >
> > Thank you for your explanation. I'm happy with most answers except the empirical results on exploitability in e.g. Leduc Poker (and also goofspiel) still seem relatively high (at ~0.2) if I understood correctly. Especially if exact BR solver is used.
> >
> > See https://arxiv.org/pdf/2201.07700.pdf Figure 4 (middle) where exploitability in this game using Double Oracle with exact BR solver is much more clearly approaching 0. While I do not expect to see faster convergence to an NE of the fully game, I do expect the algorithm to converge to a full game NE eventually at a reasonable rate; is there something I'm missing from the experimental results?

---

> > > ### Author Response · Authors · 2023-08-11
> > > **Many thanks for your quick reply.**
> > >
> > > Thanks for your quick reply! We appologize that we misunderstood the BR you mean in **Q10**.
> > > In fact, there are two "BRs".
> > > - One is the BR to the meta policy of the opponent at each iteration in PSD-PSRO training, where we use RL algorithm (PPO) to train against the opponent’s meta policy.
> > > - The other is the BR we use to evaluate the exploitability of policies. To calculate exploitability, we need a BR solver. In the definition of exploitability (Eq 1 in main paper), we have $\arg \max_{\pi'}u_i(\pi_i,\pi_{-i})=BR(\pi_{-i})$. We use an exact BR solver (available in OpenSpiel) to evaluate policies' exploitability.
> > > Hope we make it clear this time.
> > >
> > > Also, we believe the results of ~0.2 exploitability for Leduc (when approximate BRs are used) are competitively strong:
> > > - The results of SOTA methods demonstrated in Fig 3 by OpenSpeil (OpenSpiel: A Framework for Reinforcement Learning in Games, https://arxiv.org/pdf/1908.09453.pdf) use more than $10^7$ episodes to reach the average performance of ~0.3.  In comparison, PSD-PSRO use less than $3*10^6$ episodes to reach around 0.2.
> > > - Also, for the comparison to Anytime PSRO ( https://arxiv.org/pdf/2201.07700.pdf ), we should actually compared with Figure 6(a), where we believe PSD-PSRO is no worse than Anytime PSRO.

---

> > > ### Author Response · Authors · 2023-08-16
> > > **May we ask have we clarified all your confusion?**
> > >
> > > Dear Reviewer jhEH, you mentioned in your official review that  "My reservation in accepting this work is primarily due to my confusion as stated above". After our rebuttal, you followed with "I'm happy with most answers except the empirical results on exploitability"
> > >
> > > As the author-reviewer discussion phase is coming to an end, we would like to know whether our latest reply on the issue of "empirical results on exploitability" clarify your last concern.
> > >
> > > We would be happy to discuss if there is still anything unclear.
> > >
> > > Many thanks again for your time and insightful review. We really appreciate it.

---

> > > > ### Comment · Reviewer_jhEH · 2023-08-16
> > > >
> > > > Thank you for the explanation and I agree that the empirical results are as expected. I will increase my initial rating accordingly.

---

### Official Review · Reviewer_6jtZ · 2023-07-06

**Soundness:** 3 good
**Presentation:** 4 excellent
**Contribution:** 3 good
**Rating:** 7
**Confidence:** 3

**Summary:**

This paper studies how to promote policy diversity in two-player zero-sum games to better converge to a Nash Equilibrium (NE) under the algorithm framework of policy-space response oracles (PSRO). While previous works propose various diversity metrics to enlarge the gamescape, this paper shows that the enlargement of gamescape does not necessarily contribute to a better approximation of NE. Therefore, the authors propose a new diversity metric in policy space, namely KL divergence, to alleviate this issue. Theoretical results show that the proposed algorithm, PSD-PSRO, enjoys good convergence properties to NE. Experiment results in several common benchmarks for PSRO demonstrate the strong empirical performance of PSD-PSRO.

**Strengths:**

+ The main logic and presentation of this paper is very clear. Personally, I like the writing paradigm of first analyzing drawbacks of previous approaches, motivating the algorithm, and then showing strong theoretical and empirical results with the proposed algorithm.

+ This paper points out an important issue of the main-stream diversity-motivated PSRO algorithms, which in my opinion is a solid contribution to the community. To resolve this issue, the proposed algorithm is quite simple (with KL divergence) but effective.

**Weaknesses:**

+ The defined diversity metric in Eq.14 is simple in form but very complex in derivation (from line215 to line234). The fancy derivation process essentially prevents readers from obtaining the algorithm insights. I will ask detailed questions in the next section.

**Questions:**

+ I'm not quite familiar with the notation in two-player zero-sum games. In Eq.11, xi seems to be the occupancy measure over the entire state-action space. Why is the state transition probability omitted here?

+ In line221 the authors say that Eq.11 is independent of environment dynamics, but why? Eq.11 involves the trajectory \tau.

+ From my point of view, the derivation process from line215 to line234 tells the reader that KL divergence essentially measures the differernce of policies w.r.t. the sequence representation, but in my opinion this is not that straightforward. If this is generally true in RL (or two-player zero-sum games), I think there should be previous works illustrating this point as a theorem. In line230, why \beta_s can depend on the opponent policy? Depending on the opponnet policy is intuitively true, but I didn't see any theoretical evidences that it should be this way. Why does setting \beta_s in this way not changing the property that Eq.13 still measures policy differences in terms of sequence representation?



**Limitations:**

The authors have addressed several limitations in Sec.7. I didn't see any additional major limiatations.

---

> ### Author Rebuttal · Authors · 2023-08-08
>
> Thanks very much for the review and the questions. We believe that there may be some misunderstandings in L215-L234.  We list your questions below and respond to them in the following. If there is anything still unclear, please bright it up.
>
> - **Q1**: In Eq.11, $x_i$ seems to be the occupancy measure over the entire state-action space. Why is the state transition probability omitted here?
> - **Q2**: In line221 the authors say that Eq.11 is independent of environment dynamics, but why? Eq.11 involves the trajectory $\tau$.
> - **Q3**: From my point of view, the derivation process from line215 to line234 tells the reader that KL divergence essentially measures the difference of policies w.r.t. the sequence representation, but in my opinion this is not that straightforward. If this is generally true in RL (or two-player zero-sum games), I think there should be previous works illustrating this point as a theorem.
> - **Q4**: In line230, why $\beta_s$ can depend on the opponent policy? Depending on the opponent policy is intuitively true, but I didn't see any theoretical evidences that it should be this way. Why does setting \beta_s in this way not changing the property that Eq. 13 still measures policy differences in terms of sequence representation?
>
> **Responses for Q1-Q2**:
>
> $x_i$ in Eq 11 is the sequence form representation (which has been widely used for imperfect-information games [1-5] ) of a policy $\pi_i$. The joint occupancy measure is another representation of the policy $\pi_i$. The occupancy measure is a probabilistic distribution over the entire state-action space (the sequence form is **not** a probabilistic distribution). The occupancy measure of $\pi_i$ depends on $\pi_i$, the opponent policy, and the environment dynamics. In contrast, the sequence form of $\pi_i$ only depends on $\pi_i$: the opponent policy and the environment transition probability are deliberately omitted in Eq 11 when traversing a trajectory $\tau$ (because of perfect-recall, there is a unique $\tau$ that leads to (s,a)) that leads to the state-action pair (s,a). **By definition**, the right term in Eq11 only takes the product of the probability that player $i$ chooses an action along **his** infosets (states) in $\tau$, i.e., on $(\widetilde{s}_i, \widetilde{a}) \in \tau(s, a)$.
>
> Actually, our diversity metric is defined in Eq 10, which calculates a distance **between the current policy $\pi_i$ and the policy hull $\mathcal{H}(\Pi_i)$**. Section 4.2 and Appendix B.1 basically state **why&how** we encode a policy using the sequence form and how we employ the Bregman divergence on the sequence form to deduce a clean, practical, manageable, state-action based diversity objective function (Eq 14). To summarize, Eq 14 is a practical and theoretically justified implementation of Eq 10.
>
>
>
> **Responses for Q3**:
>
> As we know, Bregman Divergence is defined with a strictly convex function $d$ in Eq 16.
> When $d$ is a dilated Distance-Generating Function (DGF) [1,2] as defined in Eq 17, a recent theorem (Lemma D.2 in [3]) states that the Bregman Divergence of two policies can be transformed to a weighted sum of the Bregman divergence on the action probability distribution in a state (Eq 18).  The same theorem in simpler forms (when $d_s$ is set to a negative entropy function) have also been proven (or used) in  [4,5].  This is crucial, since it enables a practical method to calculate our Bregman Divergence **using only state-action samples**.
>
> In our work, we choose $d_s$ to be a negative entropy function (one of the dilated DGF). In this situation, $B_{d_s}$ is the KL divergence.
>
>
>
> **Responses for Q4**:
>
> According to the dilated Distance-Generating Function (Eq 17), $\beta_s >0$ is a state dependent parameter. As a result, setting $\beta_s$ as been done in Line230 can be viewed as a state dependent parameter.
>
> In line230, making $\beta_s$ depend on the opponent policy like what we did can simplify Eq 12 to Eq 13 (avoid importance sampling). Why simplification? Because we need a practical and easy way to optimize our diversity. The choices ($\beta_s$, $d_s$) above lead to a clean and practical form to calculate the distance of two polices (Eq 13). Furthermore, the gradient of Eq 14 can be deduced as in Eq 15. It is worth mentioning that when comparing two distances ($dist(\pi_a, \pi_b)$ and $dist(\pi_a, \pi_c)$), it should be based on trajectories generated by $\pi_a$ playing against the same opponent $b_{-i}$ (this leads to the same $\beta_s$).
>
> [1] Samid Hoda, Andrew Gilpin, Javier Pena, and Tuomas Sandholm. Smoothing techniques for computing nash equilibria of sequential games. Mathematics of Operations Research, 35(2):494–512, 2010.
>
> [2] Gabriele Farina, Christian Kroer, and Tuomas Sandholm. Optimistic regret minimization for extensive-form games via dilated distance-generating functions. Advances in neural information processing systems, 32, 2019.
>
> [3] Weiming Liu, Huacong Jiang, Bin Li, and Houqiang Li. Equivalence analysis between counterfactual regret minimization and online mirror descent. In Proceedings of the 39th International Conference on Machine Learning, volume 162, pages 13717–13745, 2022.
>
> [4] Lee, Chung-Wei, Christian Kroer, and Haipeng Luo. "Last-iterate convergence in extensive-form games." Advances in Neural Information Processing Systems 34 (2021): 14293-14305.
>
> [5] Bai, Yu, et al. "Near-optimal learning of extensive-form games with imperfect information." International Conference on Machine Learning. PMLR, 2022.

---

### Author Rebuttal · Authors · 2023-08-10

We provide additional experiment results in the pdf.

In the pdf, Figure 1 shows the result of continuing running on Leduc, corresponding to **Q10 by Reviewer jhEH**.

In the pdf, Figure 2 shows the result of the experiment on Goofspiel with 5 point cards (a smaller setting than the original Goofspiel (with 8 point cards) we run on in the first submission), corresponding to **Q8 by Reviewer jhEH** and **Q12 by Reviewer Mw94**.

In the pdf, Figure 3 shows the result of PSD-PSRO with different opponent settings for generating the trajectories to optimize our diversity, corresponding to **Q9 by Reviewer Mw94**.

---

### Decision · Program_Chairs · 2023-09-21

**Decision:**

Accept (poster)

**Comment:**

A case of clear acceptance with unanimous support from all reviewers. The authors are encouraged to address the comments/suggestions of the reviewers in improving the final version of their paper.